# Rethinking Disentanglement under Dependent Factors of Variation

**Antonio Almudévar**                                                                    *almudevar@unizar.es*
*ViVoLab, Aragón Institute for Engineering Research (I3A)*
*University of Zaragoza*

**Alfonso Ortega**                                                                          *ortega@unizar.es*
*ViVoLab, Aragón Institute for Engineering Research (I3A)*
*University of Zaragoza*

**Reviewed on OpenReview:** *https://openreview.net/forum?id=PgwkNC63CS*

## Abstract

Representation learning enables the discovery and extraction of underlying factors of variation from data. A representation is typically considered disentangled when it isolates these factors in a way that is interpretable to humans. Existing definitions and metrics for disentanglement often assume that the factors of variation are statistically independent. However, this assumption rarely holds in real-world settings, limiting the applicability of such definitions and metrics in real-world applications. In this work, we propose a novel definition of disentanglement grounded in information theory, which remains valid even when the factors are dependent. We show that this definition is equivalent to requiring the representation to consist of minimal and sufficient variables. Based on this formulation, we introduce a method to quantify the degree of disentanglement that remains effective in the presence of statistical dependencies among factors. Through a series of experiments, we demonstrate that our method reliably measures disentanglement in both independent and dependent settings, where existing approaches fail under the latter.

## 1 Introduction

In representation learning, data are commonly assumed to be characterized by a set of factors of variation (henceforth, factors) and nuisances. The distinction between them lies in task relevance: factors directly influence the outcome of interest, whereas nuisances do not. For instance, in a dataset of fruit images where the task is to describe the fruits, relevant factors may include fruit type, color, or size, while irrelevant nuisances might be the background color or the shadows cast. A large body of work has emphasized the importance of learning disentangled representations, where distinct dimensions of the representation correspond to different factors of variation (Schmidhuber, 1992; Bengio et al., 2013; Ridgeway, 2016; Lake et al., 2017; Achille & Soatto, 2018a). Such representations are valuable across a range of applications, including: (i) interpretability of learned features and model predictions (Hsu et al., 2017; Worrall et al., 2017; Gilpin et al., 2018; Zhang & Zhu, 2018; Liu et al., 2020; Zhu et al., 2021; Nauta et al., 2023), (ii) fairer decision-making by mitigating or removing biases in sensitive attributes such as gender or race (Calders & Žliobaitė, 2013; Barocas & Selbst, 2016; Kumar et al., 2018; Locatello et al., 2019a; Sarhan et al., 2020; Chen et al., 2023), and (iii) empowering generative models to synthesize data with controllable attributes (Yan et al., 2016; Paige et al., 2017; Huang et al., 2018; Dupont, 2018; Kazemi et al., 2019; Antoran & Miguel, 2019; Zhou et al., 2021; Shen et al., 2022; Almudévar et al., 2024).

The definition in the previous paragraph provides an intuitive, though imprecise, understanding of disentangled representations. It raises two fundamental questions: (i) what does it precisely mean to separate the factors, and (ii) how can we evaluate whether the factors are indeed separated? In Section 2, we review several works that attempt to address these two questions. Most of these approaches assume that the factors of

variation are independent—both from one another and from the nuisances. Consequently, the corresponding definitions and evaluation metrics are grounded in this assumption of independence. Many of the benchmark datasets used to validate these methods also exhibit (approximately) independent factors (LeCun et al., 2004; Liu et al., 2015; Reed et al., 2015; Matthey et al., 2017; Burgess & Kim, 2018; Gondal et al., 2019), thereby making the independence-based definitions and metrics largely applicable in those contexts.

However, in real-world scenarios, factors of variation are rarely statistically independent—either from each other or from nuisances. Returning to the earlier example, the shape of a fruit is not independent of attributes such as color, size, or shadow: encountering a yellow banana-shaped fruit is common, whereas a yellow strawberry-shaped fruit is highly unlikely. When factors are independent, separating them in a representation is relatively straightforward. In contrast, when factors are dependent, it becomes less clear what it means for a representation to "separate" them. As illustrated in Figure 1, even with a perfectly disentangled encoder, individual components of the representation can still carry information about multiple factors. In this work, we focus on defining and quantifying disentanglement in the setting where the factors of variation are not independent. Our key contributions can be summarized as follows:

- We introduce a set of four information-theoretic properties that characterize desirable behavior for disentangled representations in the presence of dependencies and nuisances.

- We prove that these properties are directly connected to the concept of *minimal* and *sufficient* representations. Moreover, we argue that evaluating minimality and sufficiency provides a more meaningful assessment of disentanglement than analyzing the properties in isolation.

- We propose a method to quantify the degree of *minimality* and *sufficiency*, enabling the evaluation of disentanglement in arbitrary representations.

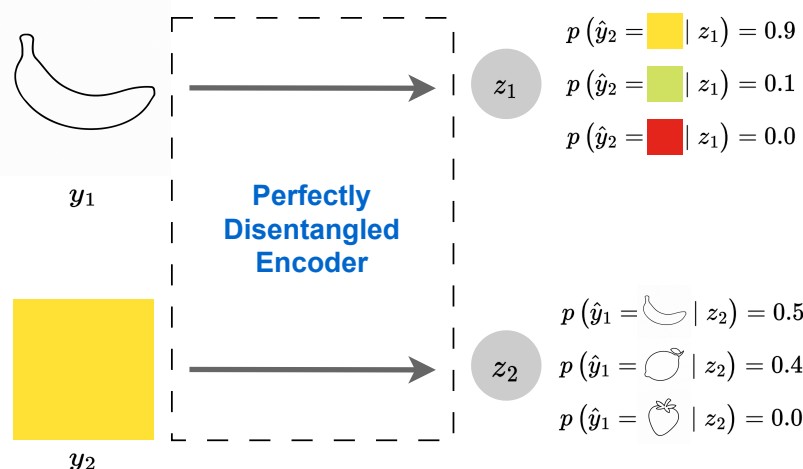

Figure 1: If the factors are dependent, then—even with a perfectly disentangled encoder—each sub-representation will contain information about other factors. For example, if $z_1$ encodes the shape of a banana, the color is most likely yellow, possibly green if unripe, and rarely red. Likewise, if $z_2$ encodes the color yellow, the shape is most likely that of a banana or a lemon, but very unlikely that of a strawberry.

## 2  Related Work

**Definition of Disentanglement**  Despite being a topic of great interest, there is no widely agreed-upon definition of disentangled representations. Intuitively, a disentangled representation separates the different factors of variation underlying the data (Desjardins et al., 2012; Bengio et al., 2013; Cohen & Welling, 2014; Kulkarni et al., 2015; Ridgeway, 2016). Early evaluations of disentanglement were largely based on visual inspection (Kingma & Welling, 2014). However, more precise definitions are necessary to understand the limitations and practical relevance of disentangled representations in downstream applications. One of the

earliest and most widely cited definitions is given in Bengio et al. (2013), and has been adopted by several subsequent works (Higgins et al., 2017; Kim & Mnih, 2018; Kim et al., 2019; Suter et al., 2019). This definition states that a disentangled representation is one in which a change in a single dimension of the representation corresponds to a change in a single factor of variation, while remaining invariant to changes in other factors. An alternative view argues that disentanglement should be defined such that a change in a single factor of variation leads to a change in only one dimension of the representation (Kumar et al., 2018; Locatello et al., 2019b). More recent efforts define disentanglement based on a set of desirable properties. Notably, Ridgeway & Mozer (2018) and Eastwood & Williams (2018) independently proposed three key properties that a representation should satisfy to be considered disentangled. While the terminology differs slightly between the two works, the underlying ideas align: (i) *Modularity* (referred to as *disentanglement* in Eastwood & Williams (2018)): each representation variable captures at most one factor of variation; (ii) *Compactness* (*completeness* in Eastwood & Williams (2018)): each factor of variation is captured by a single representation variable; and (iii) *Explicitness* (*informativeness* in Eastwood & Williams (2018)): the representation retains all task-relevant information about the factors present in the input. All of the above definitions present two key limitations: (i) they do not account for invariance to nuisance factors that may be present in the input; and (ii) they assume that the factors of variation are independent of one another and of the nuisances—which generally does not hold in real-world data. We propose a set of properties that a representation must satisfy to be considered disentangled, explicitly addressing both of these aspects.

**Metrics for Disentanglement**   Due to the lack of consensus on the definition of disentanglement, there is also no agreement on how it should be measured. Below, we provide an overview of common disentanglement metrics, organized by their underlying principles, following Carbonneau et al. (2022).

- **Intervention-based Metrics.** These methods construct data subsets that share a common factor of variation while differing in others. Representations for these subsets are computed and compared to produce a disentanglement score. Representative methods include $\beta$-VAE (Higgins et al., 2017), FactorVAE (Kim & Mnih, 2018), and R-FactorVAE (Kim et al., 2019). These metrics primarily assess modularity but do not account for compactness or explicitness (Sepliarskaia et al., 2019).

- **Information-based Metrics.** These approaches estimate mutual information between the factors and individual variables in the representation. Widely used examples include the Mutual Information Gap (MIG) (Chen et al., 2018) and Robust MIG (Do & Tran, 2020). These metrics are mainly designed to capture compactness.

- **Predictor-based Metrics.** These methods train regressors or classifiers to predict the factors from the learned representations. The resulting predictors are then analyzed to determine the contribution of each representation variable to each factor. Separated Attribute Predictability (SAP) (Kumar et al., 2018), focuses exclusively on compactness. Subsequently, more comprehensive frameworks have been proposed to evaluate multiple properties of disentanglement. Notably, Eastwood & Williams (2018) introduced DCI (Disentanglement, Completeness, and Informativeness), while Ridgeway & Mozer (2018) proposed an equivalent set: Modularity, Compactness, and Explicitness.

**Disentanglement under Dependent Factors of Variation**   As previously mentioned, disentanglement under dependent factors is a relatively underexplored area. However, several works have addressed related ideas, among which the following stand out: Suter et al. (2019) offers a causal perspective on representation learning that connects disentanglement with robustness to domain shift; Choi et al. (2021) proposes a method to disentangle variations that are shared across all classes from those that are specific to each class; Montero et al. (2020; 2022) examine the effects of excluding certain factor combinations during training and evaluate models on their ability to reconstruct unseen combinations at test time; Träuble et al. (2021) introduces synthetic dependencies between pairs of factors to study disentanglement in more realistic settings; Roth et al. (2023) presents the Hausdorff Factorized Support (HFS) criterion, which accommodates arbitrary (including correlated) distributions over factor supports; Ahuja et al. (2023) explores how interventional data can support causal representation learning by revealing geometric patterns in latent variables; and von Kügelgen et al. (2024) proposes a nonparametric method for discovering latent causal variables and their relations from high-dimensional observations, without relying on strong parametric assumptions.

**Information Bottleneck in Representation Learning**   The Information Bottleneck (IB) framework (Tishby et al., 1999) formalizes a trade-off between fidelity and compression in learned representations. A representation is deemed *sufficient* if it retains all information relevant to the target task, and *minimal* if it excludes all information irrelevant to it. Formally, given an input $X$, target variable $Y$, and representation $Z$, sufficiency corresponds to $I(Z;Y) = I(X;Y)$, while minimality requires $I(Z;X) = I(Z;Y)$. The IB principle has been leveraged in a range of works to design regularizers for representation learning (Alemi et al., 2017; 2018; Achille & Soatto, 2018b;a; Almudévar et al., 2025a), and has also been applied to promote disentanglement (Vera et al., 2018; Yamada et al., 2020; Jeon et al., 2021; Gao et al., 2021). To the best of our knowledge, however, this is the first work to formally define disentanglement in terms of minimal sufficient representations. In our formulation, each factor of variation is treated as a distinct target variable. Accordingly, we partition the overall representation vector into disjoint sets of dimensions, or 'sub-representations'. We define a representation as disentangled if each of these sub-representations is both minimal and sufficient with respect to its corresponding factor.

## 3   Desirable Properties of a Disentangled Representation

**Notation.**   We denote random variables with capital letters (e.g., $X$) and their realizations with lowercase letters (e.g., $x$). Vectors are denoted by bold lowercase letters (e.g., $\boldsymbol{x}$) and sets by calligraphic letters (e.g., $\mathcal{X}$). We use subscript indices to refer to specific components of a vector or set.

**Problem Setup.**   In our representation learning setup, we consider raw data $X \in \mathcal{X}$, which is fully explained by a set of task-relevant factors $Y = \{Y_i\}_{i=1}^n \in \mathcal{Y}$ and a set of nuisances $N \in \mathcal{N}$. The factors $Y$ correspond to the underlying sources or causes that influence the observed data and are relevant to the task at hand, whereas the nuisances $N$ refer to sources of variation in $X$ that are irrelevant to the task.

We define a representation $Z \in \mathcal{Z}$ as a random variable governed by the conditional distribution $p(\boldsymbol{z} \mid \boldsymbol{x})$. This setup induces the Markov chains $Y \leftrightarrow X \leftrightarrow Z$ and $Y_i \leftrightarrow X \leftrightarrow Z$ for all $i = 1, \ldots, n$. We further assume that $Z$ can be decomposed into a set of $m$ variables, $Z = \{Z_j\}_{j=1}^m$, where each $Z_j$ is governed by the conditional distribution $p(z_j \mid \boldsymbol{x})$.

**Desirable Properties**   Next, we define four desirable properties for a disentangled representation. These properties are conceptually related to those discussed in Section 2, but differ in a crucial aspect: while previous definitions assume independence among factors and from nuisances, our formulation explicitly accounts for dependencies. For the following definitions, we assume that $Z_j$ is the representation variable intended to describe the factor $Y_i$.

(i) **Factors-Invariance**: We say that a variable $Z_j \in Z$ is *factors-invariant* with respect to a factor $Y_i$ when it satisfies the following Markov chain:

$$\tilde{Y}_i \leftrightarrow Y_i \leftrightarrow Z_j \tag{1}$$

where $\tilde{Y}_i = \{Y_k\}_{k \neq i}$ denotes the set of all other factors. This is equivalent to the conditional independence $I(Z_j; \tilde{Y}_i \mid Y_i) = 0$, meaning that once $Y_i$ is known, $Z_j$ provides no additional information about the remaining factors $\tilde{Y}_i$. In other words, the distribution of $Z_j$ is fully determined by the value of $Y_i$ alone. This property is directly connected to modularity (Ridgeway & Mozer, 2018) (or disentanglement in Eastwood & Williams (2018)): a representation $Z_j$ is said to be modular (or disentangled) if it captures at most one factor $Y_i$. However, such definitions implicitly assume that the factors are independent. When dependencies exist among the factors—as is often the case in real-world scenarios—this definition becomes problematic. For instance, if two factors $Y_i$ and $Y_k$ are dependent, then $Y_k$ inherently contains information about $Y_i$. As a result, under the original definition, it would be impossible for $Z_j$ to simultaneously be modular and fully informative about $Y_i$, since it would necessarily encode some information about $Y_k$ as well.

(ii) **Nuisances-Invariance**: We say that a variable $Z_j \in Z$ is *nuisances-invariant* with respect to a factor $Y_i$ and a set of nuisances $N$ when it satisfies the following Markov chain:

$$N \leftrightarrow Y_i \leftrightarrow Z_j \tag{2}$$

This is equivalent to the conditional independence $I(Z_j; N \mid Y_i) = 0$, meaning that once $Y_i$ is known, $Z_j$ remains unaffected by variations in the nuisances. This property is typically neither discussed nor measured in prior work—possibly due to the difficulty of estimating the distribution of $N$. Despite this, it is an important criterion for disentangled representations. While factors-invariance ensures that $Z_j$ is unaffected by other factors $Y_k$ ($k \neq i$), we would also expect $Z_j$ to be invariant to aspects of the input that are not factors of variation—i.e., the nuisances—if the representation is truly disentangled (Carbonneau et al., 2022).

(iii) **Representations-Invariance**: We say that a variable $Z_j$ of the representation $Z$ is *representations-invariant* for a factor $Y_i$ when it satisfies the following Markov chain:

$$\tilde{Z}_j \leftrightarrow Z_j \leftrightarrow Y_i \tag{3}$$

where $\tilde{Z}_j = \{Z_k\}_{k \neq j}$ denotes all other variables in the representation. This is equivalent to the conditional independence $I(Y_i; \tilde{Z}_j \mid Z_j) = 0$, which implies that $Y_i$ can be entirely predicted from $Z_j$ alone, without any additional information from the rest of the representation. This property is useful in downstream applications: for instance, if the goal is to predict $Y_i$, one could use $Z_j$ in isolation and ignore the remaining components of $Z$. Similarly, in controllable generative models, it would suffice to manipulate $Z_j$ to control the value of $Y_i$ in the generated output $X$. This notion is closely related to compactness as defined in (Ridgeway & Mozer, 2018) (or completeness in Eastwood & Williams (2018)), where a representation $Z_j$ is said to be compact (or complete) if it is the sole component encoding information about a factor $Y_i$. As with factors-invariance, this property assumes independence among factors. In realistic scenarios where factors such as $Y_i$ and $Y_k$ are dependent, and represented by $Z_j$ and $Z_l$ respectively, it is natural for both $Z_j$ and $Z_l$ to carry information about both factors.

(iv) **Explicitness** We say that a representation $Z$ is *explicit* for a factor $Y_i$ when it satisfies the following Markov chain:

$$X \leftrightarrow Z \leftrightarrow Y_i \tag{4}$$

This means that $I(Y_i; X|Z) = 0$, i.e., $X$ provides no information about $Y_i$ when $Z$ is known or, equivalently, $Z$ contains all the information about $Y_i$. This property is referred to as explicitness in Ridgeway & Mozer (2018) and as informativeness in Eastwood & Williams (2018). Unlike the other properties discussed, explicitness is not specific to disentangled representations—it is a desirable quality for representations more generally (Bengio et al., 2013).

## 4 Measuring Disentanglement via Minimality and Sufficiency

In this section, we define minimality and sufficiency, and argue that directly measuring these properties is more convenient and informative than evaluating the four previously discussed properties independently. We also provide practical methods for estimating these quantities from data. We now present the formal definitions:

**Definition 1.** Given a representation variable $Z$ of an input $X$ and a target variable $Y$, we say that $Z$ is *minimal* with respect to $Y$ if it satisfies $I(Z; X|Y) = 0$ or, equivalently, if $I(Z; Y) = I(Z; X)$, i.e., $Z$ contains information *only* about $Y$.

**Definition 2.** Given a representation variable $Z$ of an input $X$ and a target variable $Y$, we say that $Z$ is *sufficient* with respect to $Y$ if it satisfies $I(Y; X|Z) = 0$ or, equivalently, if $I(Z; X) = I(Y; X)$, i.e., $Z$ contains *all* information about $Y$.

### 4.1 Connection between Disentanglement and Minimality and Sufficiency

We connect below the properties introduced in Section 3 to the concepts of sufficiency and minimality.

**Theorem 1.** *Let $Y = \{Y_k\}_{k=1}^n$ denote a set of factors and $Z = \{Z_j\}_{j=1}^m$ a representation. If $Z_j$ is a minimal representation of $Y_i$, it follows that $Z_j$ is factors-invariant with respect to $Y_i$ and nuisances-invariant. Equivalently, we have that:*

$$(X \leftrightarrow Y_i \leftrightarrow Z_j) \implies (\tilde{Y}_i \leftrightarrow Y_i \leftrightarrow Z_j) \wedge (N \leftrightarrow Y_i \leftrightarrow Z_j)$$

where $\tilde{Y}_i = \{Y_k\}_{k \neq i}$. Therefore, satisfying minimality implies jointly satisfying factors-invariance and nuisances-invariance. Intuitively, if $Z_j$ is fully determined by $Y_i$ (i.e., minimal), then neither the remaining factors $\tilde{Y}_i$ nor the nuisances $N$ influence $Z_j$ once $Y_i$ is known—captured by factors-invariance and nuisances-invariance, respectively. A formal proof of this theorem is provided in Appendix A.

**Theorem 2.** *Let $Y_i$ denote a factor and $Z = \{Z_k\}_{k=1}^m$ a representation. Then, $Z_j$ is a sufficient representation of $Y_i$ if and only if $Z_j$ is representations-invariant for $Y_i$ and $Z$ is explicit for $Y_i$. Equivalently, we have:* [1]

$$(X \leftrightarrow Z_j \leftrightarrow Y_i) \iff (\tilde{Z}_j \leftrightarrow Z_j \leftrightarrow Y_i) \wedge (X \leftrightarrow Z \leftrightarrow Y_i)$$

where $\tilde{Z}_j = \{Z_k\}_{k \neq j}$. Thus, satisfying sufficiency is equivalent to jointly satisfying representations-invariance and explicitness. Intuitively, if $Y_i$ is fully determined by $Z_j$ independently of $X$ (i.e., sufficiency), then there is no information about $Y_i$ in $Z$ that is not already contained in $Z_j$ (representations-invariance), and since $Z_j \in Z$, this also implies that $Y_i$ is fully captured by $Z$ (explicitness). Conversely, if all the information in $Z$ relevant to $Y_i$ is already present in $Z_j$ (representations-invariance), and $Z$ captures all information about $Y_i$ (explicitness), then $Z_j$ must fully describe $Y_i$—that is, $Z_j$ is sufficient. A formal proof of this theorem is provided in Appendix A.

## 4.2 Why Measuring Disentanglement via Minimality and Sufficiency?

As previously demonstrated, a representation is disentangled—according to the properties defined in Section 3—if and only if it is both minimal and sufficient. Next, we argue that it is more practical and principled to directly assess sufficiency and minimality, rather than evaluating the four individual properties separately.

- **Minimality vs. Factors-Invariance + Nuisances-Invariance**: When evaluating whether a representation is disentangled, the true objective is to determine whether each of its variables is influenced by a single factor—regardless of the presence of other factors or nuisances. It is not sufficient for a variable to be aligned with a specific factor if it is also significantly affected by nuisance variables. Similarly, a variable that is invariant to nuisances but still influenced by multiple factors cannot be considered disentangled. Therefore, we argue that these two properties should be assessed jointly, and—as established in Theorem 1—this can be effectively achieved by measuring minimality. Moreover, analyzing factors and nuisances separately overlooks joint effects, where correlations between remaining factors and nuisances affect $z_j$ collectively despite having no independent influence. Minimality, by contrast, captures these underlying correlations. Furthermore, from a practical standpoint, labels for nuisance variables are seldom available, which prevents the explicit calculation of nuisance-invariance.

- **Sufficiency vs. Representations-Invariance + Explicitness**: When evaluating whether a representation is disentangled, we aim to determine whether each factor can be both exclusively and fully described by a single variable in the representation. A representation in which a single variable influences a factor but captures only a small portion of its information fails to satisfy the fundamental purpose of a representation—namely, to describe the underlying factors. Conversely, if a factor is well described by the representation as a whole but is equally influenced by all variables, we cannot claim that the representation is disentangled. For this reason, we argue that it is more meaningful to assess these two aspects jointly rather than separately. As established in Theorem 2, this can be effectively achieved by measuring sufficiency.

## 4.3 Defining metrics for Minimality and Sufficiency

At the beginning of this section, we defined when a representation is minimal or sufficient. In practice, however, representations are rarely perfectly minimal or sufficient. It is therefore desirable to design continuous metrics that quantify how close a representation is to these ideals. Moreover, while minimality and sufficiency can be defined for each pair $(Z_j, Y_i)$, it is often useful to summarize these values into a single score that characterizes the entire representation $Z$. We next introduce metrics that meet these requirements.

---

[1]Note that $\iff$ denotes "if and only if", whereas $\leftrightarrow$ represents a dependency structure in a Markov chain.

**Minimality**   Recall that a variable $Z_j$ is minimal with respect to a factor $Y_i$ if $I(Z_j; X \mid Y_i) = I(Z_j; X) - I(Z_j; Y_i) = 0$. To improve interpretability and facilitate comparison across metrics, we normalize this value by $I(Z_j; X)$ to ensure it lies in $[0, 1]$. We thus define the minimality of $Z_j$ with respect to $Y_i$ as

$$m_{ij} = 1 - \frac{I(Z_j; X \mid Y_i)}{I(Z_j; X)} = \frac{I(Z_j; Y_i)}{I(Z_j; X)}. \tag{5}$$

Here, $m_{ij}$ reflects the fraction of information in $Z_j$ that is attributable to factor $Y_i$. While these values provide factor-wise insights, it is common to report a single scalar as a disentanglement score. Assuming, as in Section 3, a one-to-one correspondence between $Z_j$ and a specific $Y_i$, we define $\hat{m}_j = \max_i m_{ij}$, measuring the minimality of $Z_j$ relative to its most associated factor. The overall minimality score is then obtained by averaging across all representation variables:

$$\bar{m} = \frac{1}{m} \sum_j \hat{m}_j \tag{6}$$

**Sufficiency**   Analogously, a variable $Z_j$ is sufficient for a factor $Y_i$ if $I(Y_i; X \mid Z_j) = I(Y_i; X) - I(Y_i; Z_j) = 0$. Applying the same normalization yields the sufficiency of $Z_j$ with respect to $Y_i$:

$$s_{ij} = 1 - \frac{I(Y_i; X \mid Z_j)}{I(Y_i; X)} = \frac{I(Y_i; Z_j)}{I(Y_i; X)}. \tag{7}$$

Thus, $s_{ij}$ measures the proportion of information about $Y_i$ captured by $Z_j$. Following the same reasoning as for minimality, we compute $\hat{s}_i = \max_j s_{ij}$, which evaluates how well the most informative representation variable predicts $Y_i$. Averaging across factors yields the overall sufficiency score:

$$\bar{s} = \frac{1}{n} \sum_i \hat{s}_i \tag{8}$$

### 4.4   Computing Minimality and Sufficiency in Practice

Calculating the mutual information terms requires access to the ground-truth factor labels $Y$. Furthermore, exact computation can be intractable, particularly when the factor $Y_i$ or the representation $Z_j$ is continuous. To address this, we proceed as follows: (i) we discretize $Z_j$ by binning, a common practice in representation learning (Shwartz-Ziv & Tishby, 2017; Locatello et al., 2019b; Carbonneau et al., 2022); and (ii) in the less common case where $Y_i$ is not categorical, we also discretize it by binning, while leaving it unchanged when it is categorical. Once both $Y_i$ and $Z_j$ are categorical, we obtain: (i) $I(Z_j; Y_i)$ can be computed directly; (ii) $I(Z_j; X) = H(Z_j)$, since $Z_j$ is fully defined by $X$ and therefore $H(Z_j|X) = 0$; and (iii) $I(Y_i; X) = H(Y_i)$, by the same reasoning.

## 5   Toy Experiments

Since our goal is to evaluate whether the proposed metrics reliably capture disentanglement, we manually construct the representations in this section's experiments. This design provides full access to the ground truth and, consequently, clear expectations about the metrics' behavior.

We generate two toy experiments where a set of factors is sampled and the corresponding representations are explicitly constructed from them. This controlled setup enables a systematic comparison between our metrics and existing disentanglement metrics under conditions that are particularly challenging for prior approaches, namely: (i) dependence between factors of variation, and (ii) the presence of nuisances in the representation. Accordingly, we design two simple experiments, each addressing one of these scenarios.

### 5.1   Disentanglement under Dependence between Factors of Variation

**Definition of the factors**   We define a set of $n$ factors $\boldsymbol{y} = \{y_i\}_{i=1}^n$, where each $y_i$ is constructed from a corresponding set of latent variables $\boldsymbol{\epsilon} = \{\epsilon_i\}_{i=1}^n$, with $\epsilon_i \sim \mathcal{U}[0, 1]$ for $i = 1, \ldots, n$. Each factor is constructed

as follows:

$$y_i' = \delta\epsilon_i + \frac{1-\delta}{n-1}\sum_{l\neq i}\epsilon_l, \qquad y_i = \mathcal{B}(y_i')$$

for $i = 1, \ldots, n$, where $\delta \in \left[\frac{1}{n}, 1\right]$ is a hyperparameter that controls the level of independence among the factors, and the function $\mathcal{B} : \mathbb{R} \to \{1, 2, \ldots, K\}$ bins the continuous values into $K$ discrete classes. In particular: (i) when $\delta = 1$, each $y_i$ depends solely on $\epsilon_i$, resulting in independent factors; and (ii) when $\delta = \frac{1}{n}$, each $y_i$ is equally influenced by all elements of $\boldsymbol{\epsilon}$, yielding maximal dependence.

**Definition of the representations**   For simplicity, we define a representation $\boldsymbol{z} = \{z_j\}_{j=1}^n$ with the same number of variables as factors, where each $z_j$ is intended to correspond to $y_j$. We define each representation variable as:

$$z_j' = \alpha y_j + \frac{1-\alpha}{n-1}\sum_{i\neq j}y_i, \qquad z_j = \cos\left(\frac{\pi z_j'}{K}\right)$$

for $j = 1, \ldots, n$, where $\alpha \in \left[\frac{1}{n}, 1\right]$ is a hyperparameter that controls the degree of disentanglement. Specifically: (i) when $\alpha = 1$, each $z_j$ depends only on $y_j$, resulting in a perfectly disentangled representation; (ii) when $\alpha = \frac{1}{n}$, each $z_j$ depends equally on all factors in $\boldsymbol{y}$, yielding a fully entangled representation. The cosine function serves to introduce a non-linearity in the mapping from factors to representations. We note that, since its argument lies in $[0, \pi]$, the transformation remains bijective, ensuring that no information about the original factor is lost.

**What results should we expect?**   Under the described setup, a disentanglement metric should satisfy the following desiderata:

(i) The metric should attain a value of 1 when $\alpha = 1$, regardless of the value of $\delta$. In this case, there is a bijective mapping between each factor $y_j$ and its corresponding representation variable $z_j$, implying perfect disentanglement.

(ii) For a fixed value of $\alpha$, the metric should increase as $\delta$ decreases. Consider a scenario in which the factors are highly dependent (i.e., low $\delta$), and each representation variable $z_j$ is influenced by all factors (i.e., low $\alpha$). In such a case, most of the information in $z_j$ can still be attributed to its corresponding factor $y_j$, resulting in higher factors-invariance. This is because the other factors contributing to $z_j$ are themselves strongly correlated with $y_j$. For example, in Figure 1, suppose the encoder is not perfectly disentangled and $z_1$, intended to represent fruit shape, also encodes color. Since banana-shaped fruits are typically yellow, $z_1$ still contains information almost exclusively about shape, despite also carrying color information. Similarly, if $z_1$ does not fully capture shape because of imperfect disentanglement, the fact that it also encodes color indirectly reinforces shape information: if the fruit is yellow, its shape is much more likely to be that of a banana than that of a strawberry.

**What results do we observe?**   Figure 2 shows the results of various disentanglement metrics for $n = 4$ and $K = 5$ across different values of $\alpha$ and $\delta$. While we report results using a fixed discretization of 20 bins and 5000 data samples, we verify in Appendices B and C that the metrics' performance is largely insensitive to changes in bin counts and number of samples, respectively. We observe that *Minimality* and *Sufficiency* are the only metrics that consistently satisfy the desirable properties outlined above. More specifically, we can see that: (i) Metrics such as *MIG*, *SAP*, *Disentanglement*, and *Completeness* fail to reach a value of one when $\alpha = 1$ and $\delta = 1$, i.e., even in the fully disentangled and independent case; (ii) Metrics such as *FactorVAE*, *MIG*, *SAP*, and *Modularity* never attain their maximum values as $\delta = \frac{1}{n}$, meaning they fail to correctly reflect disentanglement when correlations are present; and (iii) *Disentanglement* and *Completeness* remain largely invariant with respect to $\delta$, contrary to the expected behavior discussed earlier.

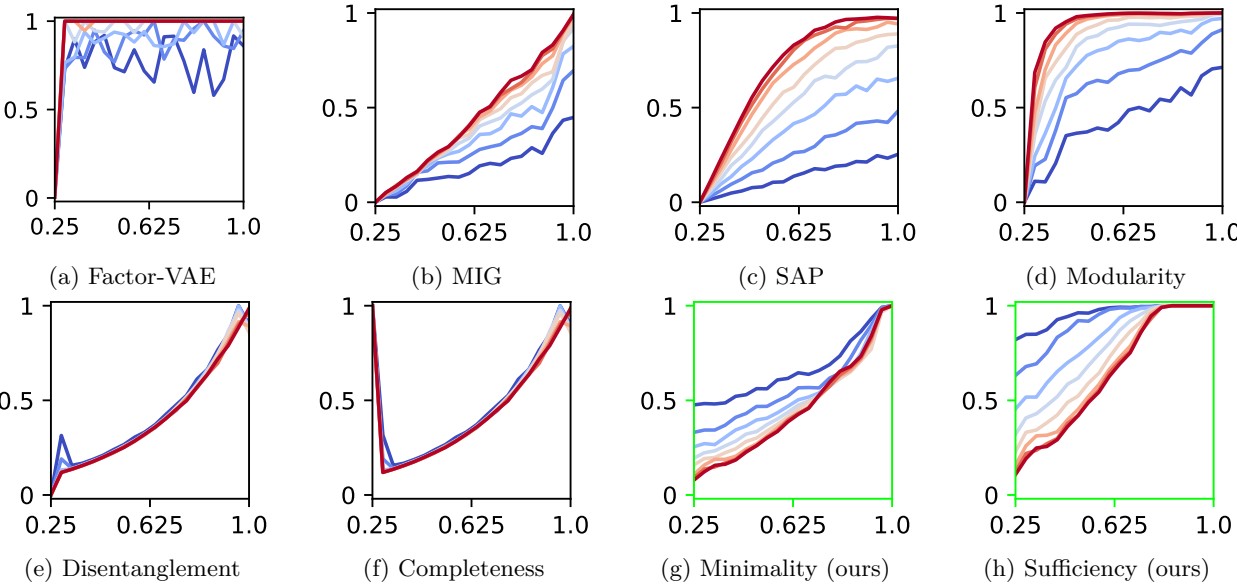

Figure 2: Comparison of different metrics (y-axis) across varying values of $\alpha$ (x-axis) and $\delta$ (color). Colors range from dark blue (highest dependence, $\delta = \frac{1}{n}$) to dark red (lowest dependence, $\delta = 1$).

## 5.2 Disentanglement under the Presence of Nuisances

**Definition of the factors** In this case, we define the factors using the same formulation as in Section 5.1, but restrict our analysis to the case $\delta = 1$, i.e., complete independence between factors.

**Definition of the representations** We define the representation $\{z_j\}_{j=1}^{n}$ such that each $z_j$ is intended to correspond to the factor $y_j$. Specifically:

$$z_j' = y_j + \beta\epsilon, \qquad z_j = \cos\left(\frac{\pi z_j'}{K}\right)$$

where $\epsilon \sim \mathcal{U}[0, 1]$ and $\beta \in \left[0, 1 - \frac{1}{K}\right]$ is a hyperparameter that controls the amount of nuisance information in the representation. As before, the argument of the cosine lies in $[0, \pi]$, ensuring that no information about $y_j$ is lost in the transformation.

**What results should we expect?** Under this setup, we expect a disentanglement metric that is sensitive to the presence of nuisances to attain its maximum value when $\beta = 0$, and to decrease as $\beta$ increases.

**What results do we observe?** Figure 3 shows how different metrics vary with respect to $\beta$. We observe that *Minimality* is the only one that is sensitive to the presence of nuisances. While our *Sufficiency* metric remains invariant to changes in $\beta$, this is the expected behavior, as shown in Theorem 2, which demonstrates that sufficiency is connected to *representations-invariance* and *explicitness*, but not to *nuisances-invariance*. Finally, we note that *Minimality* does not reach zero even at the highest value of $\beta$, because it jointly reflects both *factors-invariance* and *nuisances-invariance* and, in this setup, the former remains maximal.

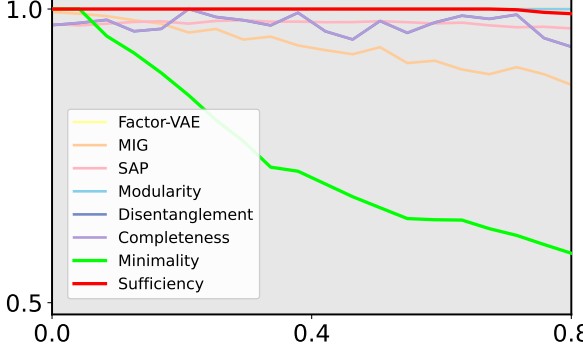

Figure 3: Comparison of different metrics (y-axis) for different values of $\beta$ (x-axis)

### 5.3 Scalability with Respect to Number of Factors

To assess the practical feasibility of these metrics in high-dimensional scenarios, we evaluate their run-time performance as the number of generative factors increases (Figure 4). Regarding the experimental setup, we clarify that *Minimality* and *Sufficiency* share a joint estimation step, similar to *Disentanglement* and *Completeness* (DCI), which is why they are displayed as pairs. The results highlight three regimes: (i) *Factor-VAE* and *SAP* are the most efficient, maintaining negligible and nearly constant execution times; (ii) *DCI* suffers from super-linear scaling costs; and (iii) our proposed metrics, while scaling almost linearly like *MIG* and *Modularity*, demonstrate superior efficiency, requiring less computation time than these linear baselines.

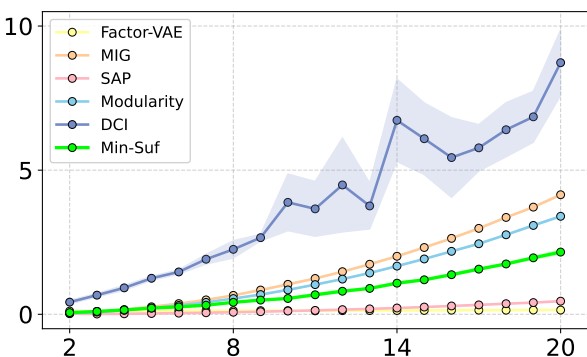

Figure 4: Time (in seconds) for calculating the different metrics (y-axis) for different number of factors of variation (x-axis).

## 6 Neural Representations Experiments

In this section, we compare our proposed metrics against existing ones on representations obtained from neural network–based encoders. Unlike in earlier experiments, ground-truth disentanglement levels are unavailable, precluding a direct analysis. Instead, as noted in Section 1, the practical value of disentangled representations often lies in their ability to enable strong performance on downstream tasks. We therefore evaluate the metrics by examining their relationship with representation quality across applications typically associated with disentanglement.

To this end, we train models known to induce disentangled representations on three widely used datasets: DSprites (Matthey et al., 2017), MPI3D (Gondal et al., 2019), and Shapes3D (Burgess & Kim, 2018). For each dataset, we explore six encoder architectures (detailed in Appendix E) combined with six disentanglement losses: $\beta$-VAE (Higgins et al., 2017), AnnealedVAE (Burgess et al., 2018), $\beta$-TCVAE (Chen et al., 2018), FactorVAE (Kim & Mnih, 2018), Ada-GVAE (Locatello et al., 2020), and HFSVAE (Roth et al., 2023), yielding 36 encoders per dataset. All models are trained using the hyperparameters listed in Appendix D.

After training, we evaluate how each metric correlates with downstream performance under varying levels of factor dependence. Following Roth et al. (2023), we introduce controlled dependencies among factors, even though the original datasets are independent by design. Specifically, we analyze cases with correlations between one, two, or three pairs of factors, as well as a confounded setting in which a single factor depends on all others. A metric suitable for dependent-factor scenarios should satisfy two criteria: (i) it should maintain a strong correlation with downstream performance, and (ii) this correlation should remain stable across different levels of factor dependence. The second condition is particularly critical: since the encoder itself does not change with the degree of factor dependence, a reliable metric should consistently reflect encoder performance irrespective of the underlying factor correlations.

### 6.1 Accuracy under Data and Computational Constraints

A central motivation for disentangled representations is their expected utility in downstream tasks, particularly under constraints of limited data and computational resources (Schölkopf et al., 2012; Bengio et al., 2013). Building on Locatello et al. (2019b), we examine whether the proposed metrics correlate with predictive performance—measured by factor prediction accuracy—under two restrictive conditions: (i) predictions rely on a single dimension of the representation, and (ii) training is performed with only 1000 samples.

**What results should we expect?** In this setting, metrics designed to assess *representations-invariance* and *explicitness* are expected to correlate strongly with predictive accuracy. The intuition is that when all the information about a factor $y_i$ is encoded within a single representation dimension $z_j$, the mapping from

$z_j$ to $y_i$ is straightforward. This implies that even a classifier with limited capacity can recover $y_i$ reliably from $z_j$, underscoring the practical value of such representations for downstream prediction.

**What results do we observe?** Figure 5 reports the rank correlation between disentanglement metrics and predictive accuracy under the low-resource setting described above. When the factors are independent, *Sufficiency* achieves the strongest correlation with accuracy, while most other metrics also exhibit moderate to high correlations. This indicates that, in the absence of dependencies, several metrics can serve as reasonable proxies for downstream performance. As factor dependence increases, however, a marked difference emerges: *Sufficiency* remains stable, whereas the correlations of the other metrics decline sharply. These findings demonstrate that *Sufficiency* provides the most reliable and robust predictor of downstream accuracy among the metrics considered, especially in scenarios where factors are dependent.

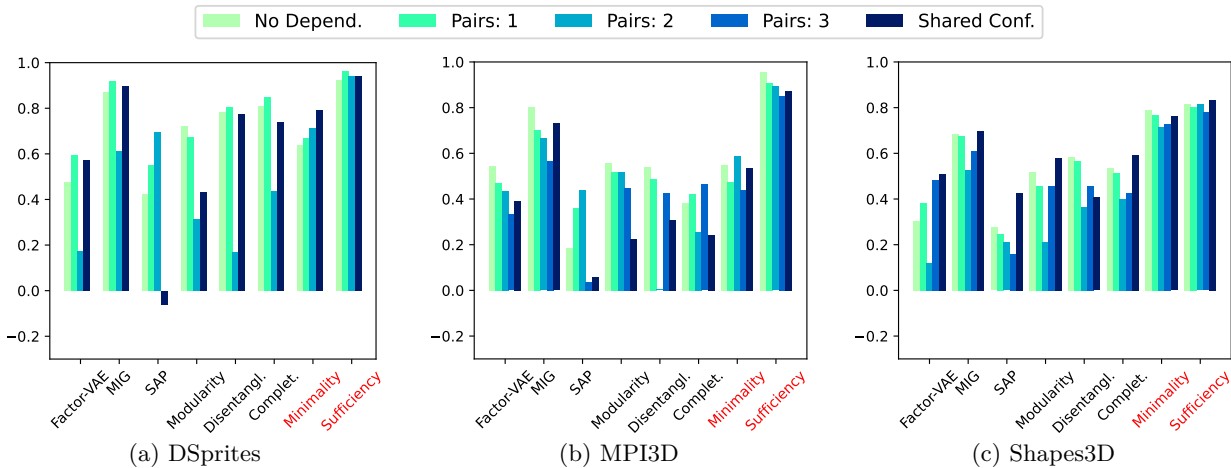

Figure 5: Rank correlation between disentanglement metrics and accuracy of a random forest classifier trained on 1000 samples across different levels of factor dependence.

## 6.2   Interventional Robustness

Another motivation for disentangled representations is their robustness to interventions on the underlying factors. Specifically, if a factor $y_i$ is modified while all others remain fixed, the resulting change in the representation should be confined to its corresponding dimension $z_j$. This property is particularly critical for controlled generative models: if it holds, modifying $z_j$ ensures that the generated data changes exclusively in the associated factor $y_i$. To quantify this, Suter et al. (2019) introduced the Interventional Robustness Score (IRS). Formally, IRS evaluates the causal link between a factor $y_i$ and a latent dimension $z_j$ by comparing post-interventional distributions. A score of 1.0 implies that $z_j$ is robust: its value changes if and only if we intervene on $y_i$ (i.e., $P(z_j|do(y_i)) \neq P(z_j|do(y_i')))$, but remains invariant if we intervene on any other factor $y_{k \neq i}$. We next compute the rank correlation between IRS and all considered metrics.

**What results should we expect?** In this setting, metrics designed to capture *factors-invariance* are expected to correlate positively with IRS. The underlying intuition is that if a representation dimension $z_j$ is entirely determined by a factor $y_i$, i.e., $p(z_j \mid y_i) = p(z_j \mid x)$, then an intervention on $y_i$—while keeping all other factors fixed—will affect $z_j$ exclusively through $y_i$. In such cases, $z_j$ can be regarded as robust to interventions, which is precisely the property that IRS is intended to measure.

**What results do we observe?** In contrast to the accuracy results, Figure 6 shows that most metrics have weak—or even negative—correlations with IRS, including when factors are independent, and their correlations vary substantially across dependence levels, sometimes changing sign. By contrast, *Minimality* exhibits a consistently strong, positive correlation with IRS across all datasets and for every dependence regime considered. The combination of stability and magnitude indicates that *Minimality* tracks interventional robustness far more faithfully than competing metrics.

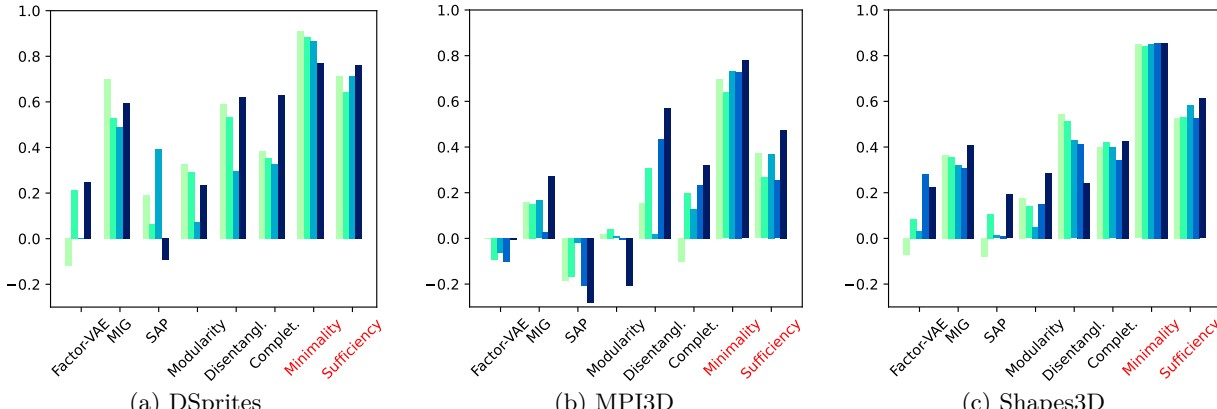

Figure 6: Rank correlation between disentanglement metrics and Interventional Robustness Score across different levels of factor dependence. Legend as in Figure 5.

## 7 Conclusions

In this work, we identified two fundamental limitations of existing definitions and metrics of disentanglement: (i) the assumption that factors of variation are independent, and (ii) the lack of consideration for invariance to nuisances in the representation. These limitations restrict the applicability of standard metrics in realistic scenarios.

To address them, we proposed four information-theoretic properties that characterize disentanglement in the presence of both dependent factors and nuisances. We showed that these properties naturally connect to the concepts of *minimality* and *sufficiency*, and argued that directly measuring these two quantities provides a more principled and interpretable characterization of disentanglement. To this end, we introduced novel metrics for quantifying *minimality* and *sufficiency*, along with tractable estimators.

Our experiments demonstrate that the proposed metrics capture disentanglement more reliably across a wider range of scenarios than existing alternatives. Moreover, unlike prior metrics, *minimality* and *sufficiency* consistently exhibit positive correlation with downstream performance in tasks typically associated with disentangled representations, even in the presence of dependent factors of variation.

## 8 Limitations and Future Directions

While our framework provides a formal and rigorous definition of disentanglement, we acknowledge two primary limitations that open avenues for future research.

**Differentiability and Optimization.** First, the proposed metrics rely on discrete binning to compute mutual information, rendering them non-differentiable. Consequently, they currently serve as evaluation tools rather than objective functions for training. A promising direction to address this is the use of variational approximations to derive differentiable upper bounds on the mutual information terms, as demonstrated in Alemi et al. (2017). Similarly, adopting an approach analogous to Almudévar et al. (2025b) could enable the direct optimization of models towards the minimality and sufficiency criteria.

**Scalability to High-Dimensional Factors.** Second, the discretization-based approximation does not scale well to scenarios where the factors of variation or representations are high-dimensional. As the dimensionality increases, the number of bins required grows exponentially, making the estimation intractable. To extend our framework to complex, high-dimensional factors, future work could leverage more robust multivariate estimators, such as the Kraskov-Stögbauer-Grassberger estimator (Kraskov et al., 2004) or the Usable Information metric (Xu et al., 2020).

**Acknowledgments**

This work has received funding from the European Union's Horizon 2020 research and innovation programme under the Marie Skłodowska-Curie grant agreement No 101007666, MCIN/AEI/10.13039/501100011033 under Grant PID2024-155948OB-C53, and the Government of Aragón (Grant Group T36 23R).

**Code Availability**

An implementation accompanying this work is available at `https://github.com/antonioalmudevar/dependent_disentanglement`

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

# A  Proofs of Theorems

**Theorem 1.** *Let $\boldsymbol{y} = \{y_l\}_{l=1}^n$ denote a set of factors and $\boldsymbol{z} = \{z_j\}_{j=1}^m$ a representation. If $z_j$ is a minimal representation of $y_i$, it follows that $z_j$ is factors-invariant with respect to $y_i$ and nuisances-invariant. Equivalently, we have that:*

$$(\boldsymbol{x} \leftrightarrow y_i \leftrightarrow z_j) \Longrightarrow (\tilde{\boldsymbol{y}}_i \leftrightarrow y_i \leftrightarrow z_j) \wedge (\boldsymbol{n} \leftrightarrow y_i \leftrightarrow z_j)$$

**Proof.** First, we demonstrate that $(\boldsymbol{x} \leftrightarrow y_i \leftrightarrow z_j) \Longrightarrow (\tilde{\boldsymbol{y}}_i \leftrightarrow y_i \leftrightarrow z_j)$: Given this Markov chain, by the DPI, we have that $I(z_j; \boldsymbol{x}) \le I(z_j; y_i)$. Since $z_j$ is a representation of $\boldsymbol{x}$, we also have by DPI that $I(z_j; \boldsymbol{y}) \le I(z_j; \boldsymbol{x})$. Therefore, it follows that $I(z_j; \boldsymbol{y}) \le I(z_j; y_i)$ On the other hand, by the chain rule of mutual information, we have that $I(z_j; \boldsymbol{y}) = I(z_j; y_i, \tilde{\boldsymbol{y}}_i) = I(z_j; y_i) + I(z_j; \tilde{\boldsymbol{y}}_i|y_i)$. By the non-negativity of mutual information, we are left with $I(z_j; \boldsymbol{y}) \ge I(z_j; y_i)$. Therefore, we have that $I(z_j; y_i) = I(z_j; \boldsymbol{y}) = I(z_j; y_i, \tilde{\boldsymbol{y}}_i)$ or, equivalently, $\tilde{\boldsymbol{y}}_i \leftrightarrow y_i \leftrightarrow z_j$.

Second, we demonstrate that $(\boldsymbol{x} \leftrightarrow y_i \leftrightarrow z_j) \Longrightarrow (\boldsymbol{n} \leftrightarrow y_i \leftrightarrow z_j)$: Given this Markov chain, we know from the DPI that $I(z_j; \boldsymbol{x}) \le I(z_j; y_i)$. On the other hand, since $z_j$ is a representation of $\boldsymbol{x}$, we have the Markov chain $(\boldsymbol{n}, y_i) \leftrightarrow \boldsymbol{x} \leftrightarrow z_j$. Equivalently, we have that $I(z_j; \boldsymbol{n}, y_i) \le I(z_j; \boldsymbol{x})$. The chain rule of mutual information tells us that $I(z_j; \boldsymbol{n}|y_i) = I(z_j; \boldsymbol{n}, y_i) - I(z_j|y_i)$. According to the above two points, we are left with $I(z_j; \boldsymbol{n}|y_i) \le I(z_j; \boldsymbol{x}) - I(z_j|y_i)$. Therefore, because of the non-negativity of mutual information, it is only possible that $I(z_j; \boldsymbol{n}|y_i) = 0$ or, equivalently, $\boldsymbol{n} \leftrightarrow y_i \leftrightarrow z_j$. $\qquad\square$

**Theorem 2.** *Let $y_i$ be a factor and $\boldsymbol{z} = \{z_l\}_{l=1}^m$ a representation. Then, $z_j$ is a sufficient representation of $y_i$ if and only if $z_j$ is representations-invariant for $y_i$ and $\boldsymbol{z}$ is explicit for $y_i$. Equivalently, we have the that:*

$$(\boldsymbol{x} \leftrightarrow z_j \leftrightarrow y_i) \Longleftrightarrow (\tilde{\boldsymbol{z}}_j \leftrightarrow z_j \leftrightarrow y_i) \wedge (\boldsymbol{x} \leftrightarrow \boldsymbol{z} \leftrightarrow y_i)$$

**Proof.** First, we demonstrate that $(\boldsymbol{x} \leftrightarrow z_j \leftrightarrow y_i) \Longrightarrow (\tilde{\boldsymbol{z}}_j \leftrightarrow z_j \leftrightarrow y_i)$: Given this Markov chain, by the DPI, we have that $I(y_i; \boldsymbol{x}) \le I(y_i; z_j)$. Since $\boldsymbol{z}$ is a representation of $\boldsymbol{x}$, we also have that $I(y_i; \boldsymbol{z}) \le I(y_i; \boldsymbol{x})$. Therefore, it follows that $I(y_i; \boldsymbol{z}) \le I(y_i; z_j)$ On the other hand, by the mutual information chain rule, we have that $I(y_i; \boldsymbol{z}) = I(y_i; z_j, \tilde{\boldsymbol{z}}_j) = I(y_i; z_j) + I(y_i; \tilde{\boldsymbol{z}}_j|z_j)$. By the non-negativity of mutual information, we are left with $I(y_i; \boldsymbol{z}) \ge I(y_i; z_j)$. Therefore, we have that $I(y_i; z_j) = I(y_i; \boldsymbol{z}) = I(y_i; z_j, \tilde{\boldsymbol{z}}_j)$ or, equivalently, $\tilde{\boldsymbol{z}}_j \leftrightarrow z_j \leftrightarrow y_i$.

Second, we demonstrate that $(\boldsymbol{x} \leftrightarrow z_j \leftrightarrow y_i) \Longrightarrow (\boldsymbol{x} \leftrightarrow \boldsymbol{z} \leftrightarrow y_i)$: Since $z_j$ is a representation of $\boldsymbol{x}$, we have that $p(y_i|\boldsymbol{x}, z_j) = p(y_i|\boldsymbol{x})$. Moreover, since $z_j$ is sufficient, we have that $p(y_i|\boldsymbol{x}, z_j) = p(y_i|z_j)$. Putting the above two terms together, we are left with $p(y_i|\boldsymbol{x}) = p(y_i|z_j)$. As we have already shown, if $z_j$ is sufficient, then we have the Markov chain $(\tilde{\boldsymbol{z}}_j \leftrightarrow z_j \leftrightarrow y_i)$, so $p(y_i|\boldsymbol{z}) = p(y_i|z_j)$. Therefore, we are left with $p(y_i|\boldsymbol{z}) = p(y_i|\boldsymbol{x})$. Finally, since $\boldsymbol{z}$ is a representation of $\boldsymbol{x}$, we know that $p(y_i|\boldsymbol{x}) = p(y_i|\boldsymbol{x}, \boldsymbol{z})$. Putting all of the above together, we are left with $p(y_i|\boldsymbol{x}, \boldsymbol{z}) = p(y_i|\boldsymbol{z})$ or, equivalently, $\boldsymbol{x} \leftrightarrow \boldsymbol{z} \leftrightarrow y_i$.

Finally, we demonstrate that $(\tilde{\boldsymbol{z}}_j \leftrightarrow z_j \leftrightarrow y_i) \wedge (\boldsymbol{x} \leftrightarrow \boldsymbol{z} \leftrightarrow y_i) \Longrightarrow (\boldsymbol{x} \leftrightarrow z_j \leftrightarrow y_i)$: Since $z_j$ and $\boldsymbol{z}$ are representations of $\boldsymbol{x}$, we have that $p(y_i|\boldsymbol{x}, z_j) = p(y_i|\boldsymbol{x}, \boldsymbol{z}) = p(y_i|\boldsymbol{x})$. Furthermore, given the Markov chain $\boldsymbol{x} \leftrightarrow \boldsymbol{z} \leftrightarrow y_i$, we know that $p(y_i|\boldsymbol{x}, \boldsymbol{z}) = p(y_i|\boldsymbol{z})$. Equivalently, given $\tilde{\boldsymbol{z}}_j \leftrightarrow z_j \leftrightarrow y_i$, we have that $p(y_i|z_j) = p(y_i|z_j, \tilde{\boldsymbol{z}}_j) = p(y_i|\boldsymbol{z})$. Recapitulating the above, we are left with $p(y_i|z_j) = p(y_i|\boldsymbol{z}) = p(y_i|\boldsymbol{x}, \boldsymbol{z}) = p(y_i|\boldsymbol{x}, z_j)$. Therefore, we have that $\boldsymbol{x} \leftrightarrow z_j \leftrightarrow y_i$. $\qquad\square$

# B    Influence of Number of Bins

## B.1    Toy Experiment 1 (Section 5.1)

As observed in Figures 7 and 8, increasing the number of bins renders our metrics less sensitive to factor correlations ($\alpha$), while still consistently satisfying the expected behavior described in Section 5.1.

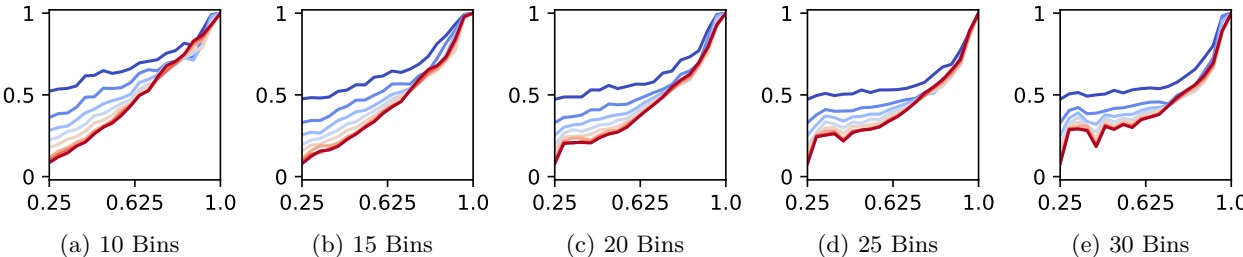

Figure 7: Comparison of Minimality for different number of bins (y-axis) across varying values of $\alpha$ (x-axis) and $\delta$ (color). Colors range from dark blue (highest dependence, $\delta = \frac{1}{n}$) to dark red (lowest dependence, $\delta = 1$).

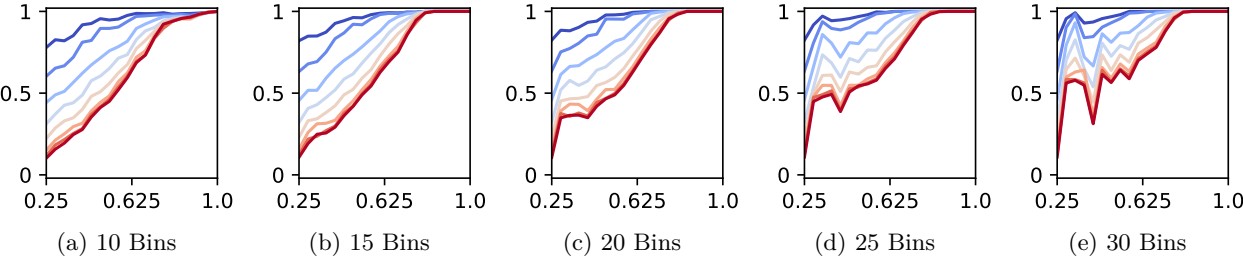

Figure 8: Comparison of Sufficiency for different number of bins (y-axis) across varying values of $\alpha$ (x-axis) and $\delta$ (color). Colors range from dark blue (highest dependence, $\delta = \frac{1}{n}$) to dark red (lowest dependence, $\delta = 1$).

## B.2    Toy Experiment 2 (Section 5.2)

In contrast to the previous section, Figure 9 reveals the opposite trend: increasing the number of bins makes *Minimality* more sensitive to $\beta$ (the amount of nuisances in the representation). Crucially, however, this increased sensitivity does not violate the expected behavior described in Section 5.2. On the other hand, *Sufficiency* remains invariant to the value of $\beta$, regardless of the number of bins used.

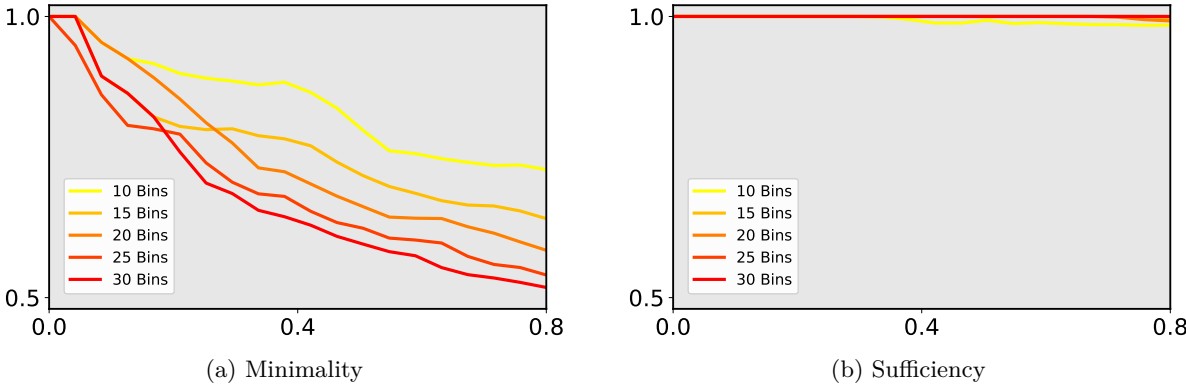

Figure 9: Comparison of Minimality and Sufficiency for different number of bins (y-axis) for different values of $\beta$ (x-axis)

### B.3 Neural Representations Experiment (Section 6)

Figures 10 and 11 demonstrate that the rank correlations for key metric pairs are robust to discretization choices. Specifically, the correlation between *Sufficiency* and accuracy under low data and computational resources, and between *Minimality* and IRS, remain almost invariant to the number of bins used for calculation.

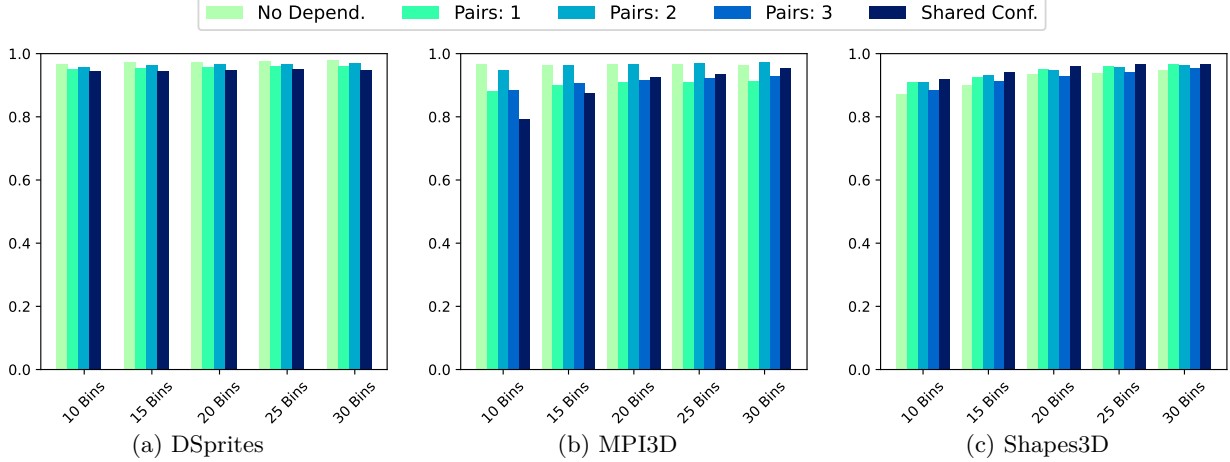

Figure 10: Rank correlation between Sufficiency for different numbers of bins and accuracy of a random forest classifier trained on 1000 samples across different levels of factor dependence.

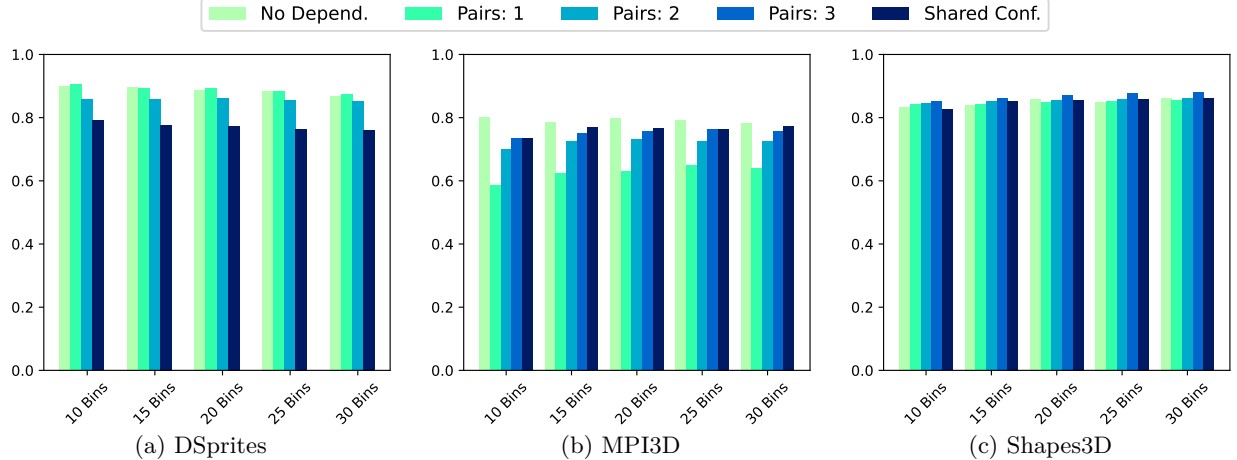

Figure 11: Rank correlation between Minimality for different numbers of bins and Interventional Robustness Score across different levels of factor dependence.

# C  Influence of the Number of Samples

## C.1  Toy Experiment 1 (Section 5.1)

As shown in Figures 12 and 13, the proposed metrics exhibit remarkable stability with respect to the number of samples, remaining consistent regardless of the factor correlations determined by $\alpha$.

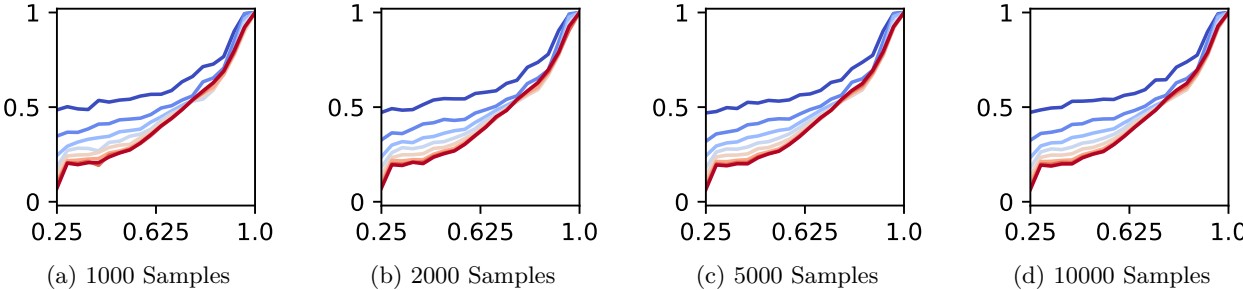

Figure 12: Comparison of Minimality for different number of samples (y-axis) across varying values of $\alpha$ (x-axis) and $\delta$ (color). Colors range from dark blue (highest dependence, $\delta = \frac{1}{n}$) to dark red (lowest dependence, $\delta = 1$).

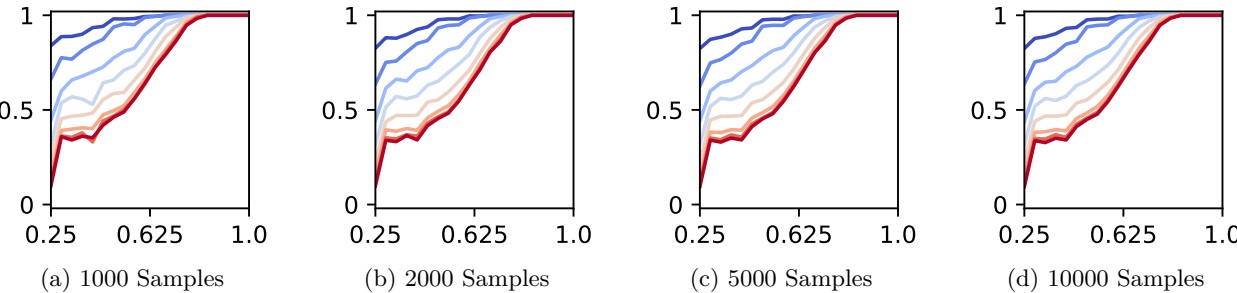

Figure 13: Comparison of Sufficiency for different number of samples (y-axis) across varying values of $\alpha$ (x-axis) and $\delta$ (color). Colors range from dark blue (highest dependence, $\delta = \frac{1}{n}$) to dark red (lowest dependence, $\delta = 1$).

## C.2  Toy Experiment 2 (Section 5.2)

As shown in Figure 14, the proposed metrics exhibit remarkable stability with respect to the number of samples, remaining consistent regardless of the level of nuisances determined by $\beta$.

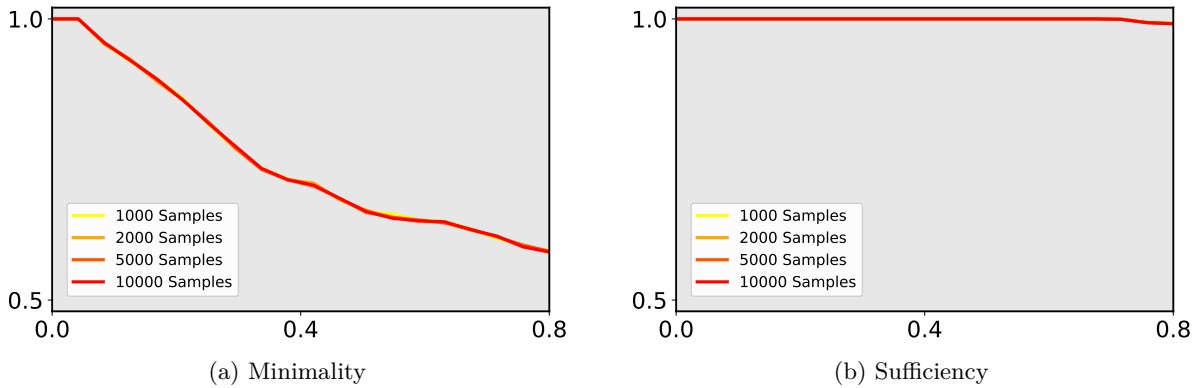

Figure 14: Comparison of Minimality and Sufficiency for different number of samples (y-axis) for different values of $\beta$ (x-axis)

### C.3 Neural Representations Experiment (Section 6)

Figures 15 and 16 demonstrate that the rank correlations for key metric pairs are robust to number of samples. Specifically, the correlation between *Sufficiency* and accuracy under low data and computational resources, and between *Minimality* and IRS, remain almost invariant to the number of samples.

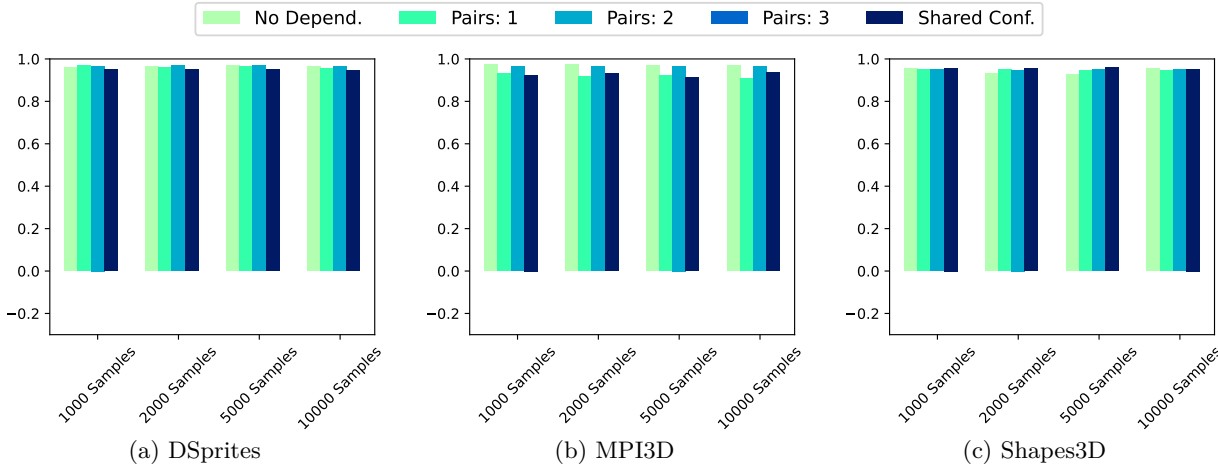

Figure 15: Rank correlation between Sufficiency for different numbers of samples and accuracy of a random forest classifier trained on 1000 samples across different levels of factor dependence.

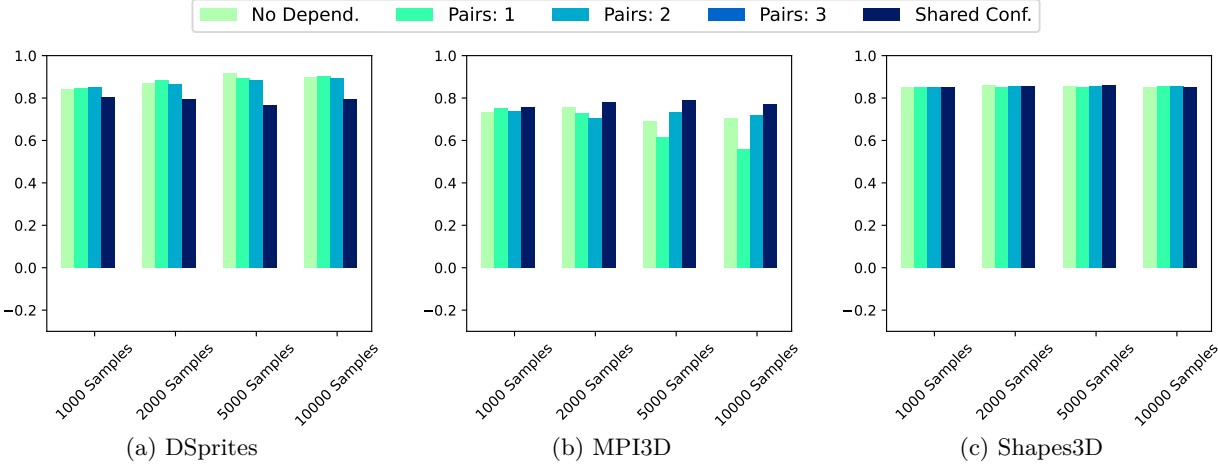

Figure 16: Rank correlation between Minimality for different numbers of samples and Interventional Robustness Score across different levels of factor dependence.

## D    Training Hyperparameters

All training hyperparameters are listed in Table 1.

Table 1: Training hyperparameters.

| Hyperparameter | Value |
|---|---|
| Batch size | 256 |
| Number of epochs | 50 |
| Optimizer | Adam |
| Base batch size | 64 |
| Learning rate | $1 \times 10^{-4}$ |
| Weight decay | $4 \times 10^{-5}$ |
| Scheduler | StepLR |
| Step size | 20 |
| Gamma | 0.1 |

## E    VAE Architectures

All our architectures are based on VAE approaches. Below, we summarize the six architectures considered, with example output shapes provided for a $64 \times 64$ RGB input image ($C{=}3$).

### E.1    Architecture Following Burgess et al. (2018)

The encoder described in Table 2 and the decoder in Table 3 are used.

Table 2: Encoder architecture (Burgess et al., 2018).

| Layer | Type | Parameters | Output shape | Activation |
|---|---|---|---|---|
| Input | – | Image size = (C, H, W) | (B, 3, 64, 64) | – |
| Conv1 | Conv2d | $C \to 32$, kernel=4, stride=2, pad=1 | (B, 32, 32, 32) | ReLU |
| Conv2 | Conv2d | $32 \to 32$, kernel=4, stride=2, pad=1 | (B, 32, 16, 16) | ReLU |
| Conv3 | Conv2d | $32 \to 32$, kernel=4, stride=2, pad=1 | (B, 32, 8, 8) | ReLU |
| Conv4* | Conv2d | $32 \to 32$, kernel=4, stride=2, pad=1 | (B, 32, 4, 4) | ReLU |
| Flatten | – | – | (B, 512) | – |
| FC1 | Linear | $512 \to 256$ | (B, 256) | ReLU |
| FC2 | Linear | $256 \to 256$ | (B, 256) | ReLU |
| FC3 | Linear | $256 \to$ latent_dim $\times 2$ | (B, latent_dim $\times 2$) | – |
| Output | Reshape | – | (B, latent_dim, 2) | – |

Table 3: Decoder architecture (Burgess et al., 2018).

| Layer | Type | Parameters | Output shape | Activation |
|---|---|---|---|---|
| Input | – | Latent vector $\in \mathbb{R}^{\text{latent\_dim}}$ | (B, latent_dim) | – |
| FC1 | Linear | latent_dim $\to 256$ | (B, 256) | ReLU |
| FC2 | Linear | $256 \to 256$ | (B, 256) | ReLU |
| FC3 | Linear | $256 \to 32 \times 4 \times 4$ | (B, 512) | ReLU |
| Reshape | – | – | (B, 32, 4, 4) | – |
| ConvT_64* | ConvTranspose2d | $32 \to 32$, kernel=4, stride=2, pad=1 | (B, 32, 8, 8) | ReLU |
| ConvT1 | ConvTranspose2d | $32 \to 32$, kernel=4, stride=2, pad=1 | (B, 32, 16, 16) | ReLU |
| ConvT2 | ConvTranspose2d | $32 \to 32$, kernel=4, stride=2, pad=1 | (B, 32, 32, 32) | ReLU |
| ConvT3 | ConvTranspose2d | $32 \to C$, kernel=4, stride=2, pad=1 | (B, C, 64, 64) | Sigmoid |

### E.2 Architecture Following Chen et al. (2018)

The encoder described in Table 4 and the decoder in Table 5 are used.

Table 4: MLP encoder architecture (Chen et al., 2018).

| Layer | Type | Parameters | Output shape | Activation |
|---|---|---|---|---|
| Input | – | Image flattened | (B, $64 \times 64 \times 3 = 12288$) | – |
| FC1 | Linear | $12288 \rightarrow 1200$ | (B, 1200) | ReLU |
| FC2 | Linear | $1200 \rightarrow 1200$ | (B, 1200) | ReLU |
| FC3 | Linear | $1200 \rightarrow$ latent_dim $\times 2$ | (B, latent_dim $\times 2$) | – |
| Output | Reshape | – | (B, latent_dim, 2) | – |

Table 5: MLP decoder architecture (Chen et al., 2018).

| Layer | Type | Parameters | Output shape | Activation |
|---|---|---|---|---|
| Input | – | Latent vector $\in \mathbb{R}^{\text{latent\_dim}}$ | (B, latent_dim) | – |
| FC1 | Linear | latent_dim $\rightarrow 1200$ | (B, 1200) | Tanh |
| FC2 | Linear | $1200 \rightarrow 1200$ | (B, 1200) | Tanh |
| FC3 | Linear | $1200 \rightarrow 1200$ | (B, 1200) | Tanh |
| FC4 | Linear | $1200 \rightarrow 64 \times 64 \times 3 = 12288$ | (B, 12288) | Sigmoid |
| Output | Reshape | – | (B, 3, 64, 64) | – |

### E.3 Architecture Following Locatello et al. (2020)

The encoder described in Table 6 and the decoder in Table 7 are used.

Table 6: Encoder architecture (Locatello et al., 2020).

| Layer | Type | Parameters | Output shape | Activation |
|---|---|---|---|---|
| Input | – | Image size = (C, 64, 64) | (B, C, 64, 64) | – |
| Conv1 | Conv2d | $C \rightarrow 32$, kernel=4, stride=2, pad=1 | (B, 32, 32, 32) | ReLU |
| Conv2 | Conv2d | $32 \rightarrow 32$, kernel=4, stride=2, pad=1 | (B, 32, 16, 16) | ReLU |
| Conv3 | Conv2d | $32 \rightarrow 64$, kernel=4, stride=2, pad=1 | (B, 64, 8, 8) | ReLU |
| Conv4 | Conv2d | $64 \rightarrow 64$, kernel=4, stride=2, pad=1 | (B, 64, 4, 4) | ReLU |
| Flatten | – | – | (B, 1024) | – |
| FC1 | Linear | $1024 \rightarrow 256$ | (B, 256) | ReLU |
| FC2 | Linear | $256 \rightarrow$ latent_dim $\times 2$ | (B, latent_dim $\times 2$) | – |
| Output | Reshape | – | (B, latent_dim, 2) | – |

Table 7: Decoder architecture (Locatello et al., 2020).

| Layer | Type | Parameters | Output shape | Activation |
|---|---|---|---|---|
| Input | – | Latent vector $\in \mathbb{R}^{\text{latent\_dim}}$ | (B, latent_dim) | – |
| FC1 | Linear | latent_dim $\rightarrow 256$ | (B, 256) | ReLU |
| FC2 | Linear | $256 \rightarrow 64 \times 4 \times 4 = 1024$ | (B, 1024) | ReLU |
| Reshape | – | – | (B, 64, 4, 4) | – |
| ConvT1 | ConvTranspose2d | $64 \rightarrow 64$, kernel=4, stride=2, pad=1 | (B, 64, 8, 8) | ReLU |
| ConvT2 | ConvTranspose2d | $64 \rightarrow 32$, kernel=4, stride=2, pad=1 | (B, 32, 16, 16) | ReLU |
| ConvT3 | ConvTranspose2d | $32 \rightarrow 32$, kernel=4, stride=2, pad=1 | (B, 32, 32, 32) | ReLU |
| ConvT4 | ConvTranspose2d | $32 \rightarrow C$, kernel=4, stride=2, pad=1 | (B, C, 64, 64) | Sigmoid |

### E.4 Architecture Combining Locatello et al. (2020) and Watters et al. (2019)

The encoder described in Table 6 and the decoder in Table 8 are used.

Table 8: Spatial Broadcast Decoder (Watters et al., 2019).

| Layer | Parameters | Output shape | Activation |
|---|---|---|---|
| Input | Latent vector | (B, latent_dim) | – |
| Spatial Broadcast | Tile latent + concat XY mesh | (B, latent_dim+2, 64, 64) | – |
| Conv1 | (latent_dim + 2) $\rightarrow$ 64, kernel=5, stride=1, pad=2 | (B, 64, 64, 64) | ReLU |
| Conv2 | 64 $\rightarrow$ 64, kernel=5, stride=1, pad=2 | (B, 64, 64, 64) | ReLU |
| Conv3 | 64 $\rightarrow$ 64, kernel=5, stride=1, pad=2 | (B, 64, 64, 64) | ReLU |
| Conv4 | 64 $\rightarrow$ 64, kernel=5, stride=1, pad=2 | (B, 64, 64, 64) | ReLU |
| Conv5 | 64 $\rightarrow$ $C$, kernel=5, stride=1, pad=2 | (B, C, 64, 64) | Sigmoid |

### E.5 Small Architecure Following Montero et al. (2022)

The encoder described in Table 9 and the decoder in Table 10 are used.

Table 9: Small encoder architecture (Montero et al., 2022).

| Layer | Type | Parameters | Output shape | Activation |
|---|---|---|---|---|
| Input | – | Image size = (C, 64, 64) | (B, C, 64, 64) | – |
| Conv1 | Conv2d | $C \rightarrow$ 32, kernel=4, stride=2, pad=1 | (B, 32, 32, 32) | ReLU |
| Conv2 | Conv2d | 32 $\rightarrow$ 32, kernel=4, stride=2, pad=1 | (B, 32, 16, 16) | ReLU |
| Conv3 | Conv2d | 32 $\rightarrow$ 64, kernel=4, stride=2, pad=1 | (B, 64, 8, 8) | ReLU |
| Conv4 | Conv2d | 64 $\rightarrow$ 64, kernel=4, stride=2, pad=1 | (B, 64, 4, 4) | ReLU |
| Conv5 | Conv2d | 64 $\rightarrow$ 128, kernel=4, stride=2, pad=1 | (B, 128, 2, 2) | ReLU |
| Flatten | – | – | (B, 512) | – |
| FC1 | Linear | 512 $\rightarrow$ 256 | (B, 256) | ReLU |
| FC2 | Linear | 256 $\rightarrow$ latent_dim $\times$ 2 | (B, latent_dim $\times$ 2) | – |
| Output | Reshape | – | (B, latent_dim, 2) | – |

Table 10: Small decoder architecture (Montero et al., 2022).

| Layer | Type | Parameters | Output shape | Activation |
|---|---|---|---|---|
| Input | – | Latent vector $\in \mathbb{R}^{\text{latent\_dim}}$ | (B, latent_dim) | – |
| FC1 | Linear | latent_dim $\rightarrow$ 256 | (B, 256) | ReLU |
| FC2 | Linear | 256 $\rightarrow$ 128 $\times$ 2 $\times$ 2 = 512 | (B, 512) | ReLU |
| Reshape | – | – | (B, 128, 2, 2) | – |
| ConvT1 | ConvTranspose2d | 128 $\rightarrow$ 64, kernel=4, stride=2, pad=1 | (B, 64, 4, 4) | ReLU |
| ConvT2 | ConvTranspose2d | 64 $\rightarrow$ 64, kernel=4, stride=2, pad=1 | (B, 64, 8, 8) | ReLU |
| ConvT3 | ConvTranspose2d | 64 $\rightarrow$ 32, kernel=4, stride=2, pad=1 | (B, 32, 16, 16) | ReLU |
| ConvT4 | ConvTranspose2d | 32 $\rightarrow$ 32, kernel=4, stride=2, pad=1 | (B, 32, 32, 32) | ReLU |
| ConvT5 | ConvTranspose2d | 32 $\rightarrow$ $C$, kernel=4, stride=2, pad=1 | (B, C, 64, 64) | Sigmoid |

### E.6 Large Architecure Following Montero et al. (2022)

The encoder described in Table 11 and the decoder in Table 12 are used.

Table 11: Large encoder architecture (Montero et al., 2022)).

| Layer | Type | Parameters | Output shape | Activation |
|---|---|---|---|---|
| Input | – | Image size = (C, 64, 64) | (B, C, 64, 64) | – |
| Conv1 | Conv2d | $C \to 64$, kernel=4, stride=2, pad=1 | (B, 64, 32, 32) | ReLU |
| Conv2 | Conv2d | $64 \to 64$, kernel=4, stride=2, pad=1 | (B, 64, 16, 16) | ReLU |
| Conv3 | Conv2d | $64 \to 128$, kernel=4, stride=2, pad=1 | (B, 128, 8, 8) | ReLU |
| Conv4 | Conv2d | $128 \to 128$, kernel=4, stride=2, pad=1 | (B, 128, 4, 4) | ReLU |
| Conv5 | Conv2d | $128 \to 256$, kernel=4, stride=2, pad=1 | (B, 256, 2, 2) | ReLU |
| Flatten | – | – | (B, 1024) | – |
| FC1 | Linear | $1024 \to 256$ | (B, 256) | ReLU |
| FC2 | Linear | $256 \to \text{latent\_dim} \times 2$ | (B, latent\_dim $\times$ 2) | – |
| Output | Reshape | – | (B, latent\_dim, 2) | – |

Table 12: Large decoder architecture (Montero et al., 2022).

| Layer | Type | Parameters | Output shape | Activation |
|---|---|---|---|---|
| Input | – | Latent vector $\in \mathbb{R}^{\text{latent\_dim}}$ | (B, latent\_dim) | – |
| FC1 | Linear | $\text{latent\_dim} \to 256$ | (B, 256) | ReLU |
| FC2 | Linear | $256 \to 256 \times 2 \times 2 = 1024$ | (B, 1024) | ReLU |
| Reshape | – | – | (B, 256, 2, 2) | – |
| ConvT1 | ConvTranspose2d | $256 \to 128$, kernel=4, stride=2, pad=1 | (B, 128, 4, 4) | ReLU |
| ConvT2 | ConvTranspose2d | $128 \to 128$, kernel=4, stride=2, pad=1 | (B, 128, 8, 8) | ReLU |
| ConvT3 | ConvTranspose2d | $128 \to 64$, kernel=4, stride=2, pad=1 | (B, 64, 16, 16) | ReLU |
| ConvT4 | ConvTranspose2d | $64 \to 64$, kernel=4, stride=2, pad=1 | (B, 64, 32, 32) | ReLU |
| ConvT5 | ConvTranspose2d | $64 \to C$, kernel=4, stride=2, pad=1 | (B, C, 64, 64) | Sigmoid |

