# OpenReview forum: "Rethinking Disentanglement under Dependent Factors of Variation"
_TMLR — Accepted by TMLR_

### Review · Reviewer_vKVM · 2025-10-20

**Summary Of Contributions:**

**Attention!** This article is outside of my main research area.

---

This paper gives a definition of disentanglement for the case where the true latent variables (or factors) are correlated and the data contain irrelevant noise (nuisances). First, it shows that existing definitions of disentanglement are not suitable for such realistic cases, since they assume the ground-truth latents are independent. Then, it introduces four desirable properties of disentangled representations: factors-invariance, nuisances-invariance, representation-invariance, and explicitness. Each of these is formulated through conditional independencies among the data, latent variables, and noise. After that, the paper demonstrates that these four properties can be expressed equivalently through two more intuitive ones: minimality and sufficiency. For both minimality and sufficiency, the paper proposes normalized metrics that make them measurable in practice. Finally, it compares the new metrics with existing ones using two toy examples and three neural representation learning datasets, showing that the proposed metrics capture disentanglement more effectively.

Strengths: The paper makes an interesting contribution and is clearly written. The experiments fulfill their purpose of comparing the proposed metrics with existing ones. The proofs appear correct.

Weaknesses: The paper could be more self-contained.

**Additional Comments:**

- $x$, $y$, and $z$ should be bold in the paragraph “Information Bottleneck in Representation Learning” on page 4.
- Both $l$ and $k$ are used for defining tilde variables; it seems possible to avoid $l$.
- A space is missing between “$y_i$” and “is” in section 4.4.
- $x$ should be bold in the expression “Thus, $p(z_j \mid x)$ …” on page 17.
- On page 17, Bayes’ theorem should read $p(z_j, \mathbf{y}, \mathbf{n}) = \ldots = p(\mathbf{n}, z_j, \mathbf{y})p(z_j \mid \mathbf{y})p(\mathbf{y})$, not $p(z_j, \mathbf{y}, \mathbf{n}) = \ldots = p(\mathbf{n}, z_j, \mathbf{y})p(z_j \mid y_i)p(\mathbf{y})$.

**Audience:**

Yes

**Audience Explanation:**

Learning disentangled representations is relevant for many applications. The contributions of this paper are useful for practical use of such representations.

**Broader Impact Concerns:**

This is a theoretical work that proposes a novel definition of disentanglement. I have no broader impact concerns.

**Claims And Evidence:**

Yes

**Claims Explanation:**

- The problems with existing definitions of disentanglement are well motivated in the paper.
- The paper includes proofs of its theoretical results in the appendix, and these proofs are correct.
- The experiments clearly show the advantages of the proposed disentanglement metrics over the existing ones.

**Requested Changes:**

- Footnote 1 contains a dead link. (critical)
- To make the paper more self-contained, briefly and formally define the most important existing definitions.
- Add a short preliminaries section summarizing the notation used.
- The term “sub-representations” on page 4 is unclear. Please clarify. (critical)
- State the number of bins in the main text rather than in a footnote, as done for $N$ and $K$.
- In Figure 3, the x-axis is clipped for $K=5$ since $1 - 1/K = 0.8$. Please correct this. (critical)
- Include a brief description of the encoder architectures used in the main text (page 10).
- Provide a formal definition of the interventional robustness score.
- Some cited arXiv papers have published versions. Please update the references. (critical)
- The third part of the proof of Theorem 1 is difficult to follow. Please rewrite it for clarity. (critical)

---

> ### Comment · Reviewer_vKVM · 2025-12-15
> **All points addressed**
>
> Authors have addressed all bigger points in my review. There are still some presentation issues. For example: Often VAE is not capitalized in Bibliography.

---

### Review · Reviewer_2H2y · 2025-11-23

**Summary Of Contributions:**

The paper studies disentangled representations in settings where **generative factors are statistically dependent and nuisances are present**, arguing that many popular metrics (MIG, SAP, DCI, etc.) are calibrated to independence-heavy benchmarks and can behave unreliably in these regimes.
It introduces **four information-theoretic properties** (factors-invariance, nuisances-invariance, representations-invariance, and explicitness) and shows that, taken together, they are equivalent to each coordinate being a minimal and sufficient representation of a single factor.
Based on this, the authors define **two concrete metrics** implemented via mutual information estimates from discretized codes and factors.
Through synthetic experiments with controlled factor dependence and nuisance strength, they report that sufficiency correlates best and most robustly with downstream predictive performance, while minimality tracks an existing interventional robustness score more reliably than other disentanglement metrics.

**Additional Comments:**

I appreciate that the paper starts from clearly stated information-theoretic properties before introducing metrics, rather than proposing a score first and only giving informal intuition later.
The decision to separate minimality and sufficiency, instead of collapsing everything into a single "disentanglement score", also makes the results more interpretable.

**Audience:**

Yes

**Audience Explanation:**

The paper addresses a real and currently under-served niche: how to evaluate disentangled representations when factors are dependent and nuisances are present, instead of assuming the independent factor, fully-observed regimes of standard benchmarks.
This is clearly relevant to parts of the TMLR audience working on disentangled/causal representation learning, identifiability of representations, and evaluation methodology.

The idea of starting from logical/information-theoretic properties and then deriving concrete quantitative metrics also connects to a growing line of work on principled disentanglement evaluation, as well as to recent efforts on more realistic disentanglement/causal representation learning setups.

- Cian Eastwood, et al. "DCI-ES: An Extended Disentanglement Framework with Connections to Identifiability." International Conference on Learning Representations. 2023.
- Yivan Zhang, Masashi Sugiyama. "Enriching Disentanglement: From Logical Definitions to Quantitative Metrics." Neural Information Processing Systems. 2024.
- Sébastien Lachapelle, et al. "Nonparametric Partial Disentanglement via Mechanism Sparsity: Sparse Actions, Interventions and Sparse Temporal Dependencies." arXiv:2401.04890. 2024.
- Dingling Yao, et al. "Multi-View Causal Representation Learning with Partial Observability." International Conference on Learning Representations. 2024.

**Claims And Evidence:**

Yes

**Claims Explanation:**

The core positive claims are reasonably well supported within the scope of the paper. The authors clearly specify their four information-theoretic properties, derive the minimality and sufficiency scores from mutual information identities, and then design synthetic experiments that systematically vary disentanglement strength, factor dependence, and nuisance strength. I find the evidence adequate.

However, some of the negative claims about prior work feel more rhetorical than demonstrated.
The paper states that existing definitions
> (i) they do not account for invariance to nuisance factors that may be present in the input; and
> (ii) they assume that the factors of variation are independent of one another and of the nuisances—which generally does not hold in real-world data.

but this is asserted rather than carefully proved.
Classical notions such as modularity/compactness/explicitness are mostly structural properties of the mapping $Y \to Z$ from factors to representations; they are typically defined without explicitly fixing which factors are "nuisances" or requiring independence of the factor distribution.

`Bengio et al. (2013)` stated that
> the most robust approach to feature learning is to disentangle as many factors as possible, discarding as little information about the data as is practical

so "nuisance factors" may be useful in some other tasks.
Further, the definitions given in some previous works are rather vague.
If the author wants to criticize the existing definitions or metrics, they should first give a clear mathematical formulation that best aligns with the original papers.
It would strengthen the paper if the authors either (i) softened this rhetoric and framed their contribution as extending these notions to explicitly model nuisances and dependent factors, or (ii) provided concrete counterexamples showing a representation that is intuitively disentangled in their sense but necessarily scored as "bad" by previous definitions/metrics.

Finally, the support for the proposed metrics themselves is somewhat limited by implementation choices.
Minimality and sufficiency are estimated via discretization and histogram-based mutual information, but there is no robustness study on sensitivity to bin counts, sample size, or distributional shape.
Since the central contribution is an evaluation metric, I would have liked at least a brief analysis or ablation here.
Moreover, all empirical evidence is on synthetic data with known factors and nuisances; this is appropriate for an initial study, but it means that stronger claims about real-world applicability should be read as suggestive rather than definitively established.

**Requested Changes:**

### Clarify the formal status of the metrics

Since the main contribution is an evaluation metric, I would like to see basic theoretical guarantees for minimality and sufficiency.
For example, explicit propositions of the form "the minimality score for $(z_j, y_i)$ equals $1$ if and only if factors-invariance and nuisances-invariance hold" etc.
Right now these relationships are described informally; turning them into clear metric-property statements would make the framework more principled and easier to interpret.

### Discuss differentiability, optimisation, and computational cost

The current implementation uses discretization and histogram-based mutual information, which is fine for post-hoc evaluation but not directly usable as a training objective.
It would improve the paper to comment on whether differentiable estimators of the same quantities are feasible, and whether in principle one could optimize disentanglement directly with these scores as losses.
Relatedly, a brief discussion of computational complexity and practical scaling (in terms of number of factors, representation dimensions, and samples) would help readers understand when the metrics are usable in practice.

### Be explicit about supervision and data requirements

The experiments assume access to ground-truth factors (and nuisances) to estimate mutual information.
It would be useful to spell out what is required in general: Do we need labels for all factors and nuisances? Can the metrics be computed with only a subset of factors observed? How do they behave under strongly imbalanced factor distributions? Even if only discussed qualitatively, this would clarify the scope of applicability.

---

### Review · Reviewer_AQT3 · 2026-01-05

**Summary Of Contributions:**

COPYING MY COMMENT AS A REVIEW

The authors are focusing in disentangled representations, in the setup where latent factors are dependent and nuisances appear. Drawing from the current stage of popular metrics e.g. mIG, SAP,DCI that assume independent generative factors, the authors challenge the metrics' ability to represent real-wolrd settings. For that they propose 4 information-theoretic properties that a disentangled representaton should satisfy. They run toy experiments, they construct settings with increasing nuisance information to assess which metrics react appropriately. Overall, Sufficiency tends to correlate best and most robustly with downstream accuracy as dependence increases, and Minimality correlates most reliably with IRS.

Pros:

The motivation is clear. The paper does a good job on presenting the problem the independent factor assumption.
The synthetic experiments seem reasonable. The variability of the disentaglement strength , factor dependence, and nuisance injection is well defined.
I like the way the authors break disentanglement down into explicit information-theoretic properties. It's an intuitive way to tie them back to sufficinet representations.
The correlation analysis with low-data prediction and IRS is convincing.
The paper is clearly written and straightforward to follow.
Cons:

Although some of these assumptions that the authors make are standard in disentanglement benchmarks, the conditional independence of factors given
 one seems unusual to me and the authors should use more justification and discussion. It would help to make clear which results really depend on it, and whether any weaker assumptions would suffice.
the metrics are relying on discretizing
 and sometimes
, and then on using histogram-based MI. I'm concerned that this choice might be sensitive to a) the numer of the bins b) sample size, c) and the distributional shape of the varialbes. Since the contribution is an evaluation metric, I would really like to see at least a brief robustness study here.
The main link—disentanglement as minimal and sufficient representations in the information-bottleneck sense—has been touched on in prior work, so the main novelty of the paper defining concrete evaluation metrics that make these connections concise. I think this is still a meaningful novelty, but the paper might want to present itself more as “a solid and principled evaluation framework” rather than suggesting a completely new theory of disentanglement.
Questions:

Can you calrify how siginificant is the conditional independence $p(y_i, y_j | x) = p(y_i| x) p(y_j|x) ?
Possibly a moving-forward question on the possibility for distributed representations, rather than concern: Do you see a way to extend your framework to handle the case where a factor is represented by a group of coordinates rather than a single one?
How sensitive are the Minimality and Sufficiency scores to the binning and sample size? Have you tried different bin counts or alternative MI estimators?
Overall, I'm leaning positive toward this submission: the paper discusses a real gap in how the community evaluates disentanglement when factors are dependent and nuisances are present, and the information-theoretic formulation is clean.

**Audience:**

Yes

**Audience Explanation:**

Explained above

**Broader Impact Concerns:**

Explained above

**Claims And Evidence:**

Yes

**Claims Explanation:**

Explained above

**Requested Changes:**

Mentioned above

---

> ### Author Response · Authors · 2026-01-09
> **Response to Reviewer AQT3**
>
> We copy our previous answer below
>
> ---
>
> We thank the reviewer for their thoughtful feedback and constructive comments. Below, we provide a point-by-point response to the concerns and questions raised.
>
> ### Cons
>
> > 1. Although some of these assumptions that the authors make are standard in disentanglement benchmarks, the conditional independence of factors given $x$ one seems unusual to me and the authors should use more justification and discussion. It would help to make clear which results really depend on it, and whether any weaker assumptions would suffice.
>
> We appreciate the reviewer for pointing this out. After re-examining our proofs, we confirmed that the assumption of conditional independence of factors given $x$ is not required for the validity of Theorems 1 and 2. Accordingly, we have removed this assumption from the manuscript, which we believe simplifies the framework and enhances the generality of our findings.
>
> > 2. the metrics are relying on **discretizing** $z_j$ and sometimes $y_i$, and then on using histogram-based MI. I'm concerned that this choice might be sensitive to a) the numer of the bins b) sample size, c) and the distributional shape of the varialbes. Since the contribution is an evaluation metric, I would really like to see at least a brief robustness study here.
>
> We agree that establishing robustness is crucial for any proposed metric. To this end, we have expanded our experimental analysis to evaluate sensitivity regarding both bin counts and sample size:
>
> - **Bin Counts:** We have included an ablation study showing that the results remain consistent across different numbers of bins (Appendix B).
> - **Sample Size:** We have demonstrated that the dataset size has very little effect on our metrics (Appendix C).
>
> Regarding the concern about **distributional shape**, we are not entirely certain which specific analysis is being requested. We would be grateful for further clarification on this point and are more than willing to conduct the necessary analysis once we have more details.
>
> > 3. The main link—disentanglement as minimal and sufficient representations in the information-bottleneck sense—has been touched on in prior work, so the main novelty of the paper defining concrete evaluation metrics that make these connections concise. I think this is still a meaningful novelty, but the paper might want to present itself more as “a solid and principled evaluation framework” rather than suggesting a completely new theory of disentanglement.
>
> We are not currently aware of specific prior work that explicitly links disentanglement to minimal sufficient representations in the manner we describe. We would be very interested in receiving these references so that we can properly cite and discuss them in our paper.
>
> Regarding the positioning of our contribution, we fully agree that our work should be viewed as a principled evaluation framework rather than a completely new theory of disentanglement. If there are specific sections that suggest a broader theoretical claim, we would appreciate it if you could point them out so we can clarify the text to better reflect the scope of our contribution.
>
> ---
>
> ### Questions
>
> > 1. Can you calrify how siginificant is the conditional independence $p(y_i, y_j | x) = p(y_i| x) p(y_j|x)$?
>
> Please refer to our response to Con 1, where we address the removal of this assumption.
>
> > 2. Possibly a moving-forward question on the possibility for distributed representations, rather than concern: Do you see a way to extend your framework to handle the case where a factor is represented by a group of coordinates rather than a single one?
>
> We agree that extending our framework to handle distributed representations—where factors correspond to groups of coordinates—is an important avenue for future research. We have included a discussion in the "Future Work" section detailing how multivariate estimators, such as KSG or $\mathcal{V}$-Information, could be utilized to measure Minimality and Sufficiency on subspaces rather than single coordinates.
>
> > 3. How sensitive are the Minimality and Sufficiency scores to the binning and sample size? Have you tried different bin counts or alternative MI estimators?
>
> Please refer to our response to Con 2, where we discuss our robustness analysis regarding binning and sample size.
>
> ---
>
> We hope that these responses and the corresponding revisions satisfactorily address the reviewer's concerns. We remain open to any further questions or suggestions.

---

### Comment · Reviewer_AQT3 · 2025-11-18
**Review for Rethinking Disentanglement under Dependent Factors of Variation**

The authors are focusing in disentangled representations, in the setup where latent factors are dependent and nuisances appear. Drawing from the current stage of popular metrics e.g. mIG, SAP,DCI that assume independent generative factors, the authors challenge the metrics' ability to represent real-wolrd settings. For that they propose 4 information-theoretic properties that a disentangled representaton should satisfy. They run toy experiments, they construct settings with increasing nuisance information to assess which metrics react appropriately. Overall, Sufficiency tends to correlate best and most robustly with downstream accuracy as dependence increases, and Minimality correlates most reliably with IRS.

*Pros*:
1. The motivation is clear. The paper does a good job on presenting the problem the independent factor assumption.
2. The synthetic experiments seem reasonable. The variability of the disentaglement strength , factor dependence, and nuisance injection is well defined.
3. I like the way the authors break disentanglement down into explicit information-theoretic properties. It's an intuitive way to tie them back to sufficinet representations.
4. The correlation analysis with low-data prediction and IRS is convincing.
5. The paper is clearly written and straightforward to follow.

*Cons*:
1. Although some of these assumptions that the authors make are standard in disentanglement benchmarks, the conditional independence of factors given $x$ one seems unusual to me and the authors should use more justification and discussion. It would help to make clear which results really depend on it, and whether any weaker assumptions would suffice.
2. the metrics are relying on **discretizing** $z_j$ and sometimes $y_i$, and then on using **histogram-based** MI. I'm concerned that this choice might be sensitive to a) the numer of the bins b) sample size, c) and the distributional shape of the varialbes. Since the contribution is an evaluation metric, I would really like to see at least a brief robustness study here.
3. The main link—disentanglement as minimal and sufficient representations in the information-bottleneck sense—has been touched on in prior work, so the main *novelty* of the paper defining concrete evaluation metrics that make these connections concise. I think this is still a meaningful novelty, but the paper might want to present itself more as “a solid and principled evaluation framework” rather than suggesting a completely new theory of disentanglement.

*Questions*:
1. Can you calrify how siginificant is the conditional independence $p(y_i, y_j | x) = p(y_i| x) p(y_j|x) ?
2. Possibly a moving-forward question on the possibility for distributed representations, rather than concern: Do you see a way to extend your framework to handle the case where a factor is represented by a group of coordinates rather than a single one?
3. How sensitive are the Minimality and Sufficiency scores to the binning and sample size? Have you tried different bin counts or alternative MI estimators?

Overall, I'm leaning positive toward this submission: the paper discusses a real gap in how the community evaluates disentanglement when factors are dependent and nuisances are present, and the information-theoretic formulation is clean.

---

> ### Author Response · Authors · 2025-12-03
> **Response to Reviewer AQT3**
>
> We thank the reviewer for their thoughtful feedback and constructive comments. Below, we provide a point-by-point response to the concerns and questions raised.
>
> ### Cons
>
> > 1. Although some of these assumptions that the authors make are standard in disentanglement benchmarks, the conditional independence of factors given $x$ one seems unusual to me and the authors should use more justification and discussion. It would help to make clear which results really depend on it, and whether any weaker assumptions would suffice.
>
> We appreciate the reviewer for pointing this out. After re-examining our proofs, we confirmed that the assumption of conditional independence of factors given $x$ is not required for the validity of Theorems 1 and 2. Accordingly, we have removed this assumption from the manuscript, which we believe simplifies the framework and enhances the generality of our findings.
>
> > 2. the metrics are relying on **discretizing** $z_j$ and sometimes $y_i$, and then on using histogram-based MI. I'm concerned that this choice might be sensitive to a) the numer of the bins b) sample size, c) and the distributional shape of the varialbes. Since the contribution is an evaluation metric, I would really like to see at least a brief robustness study here.
>
> We agree that establishing robustness is crucial for any proposed metric. To this end, we have expanded our experimental analysis to evaluate sensitivity regarding both bin counts and sample size:
>
> - **Bin Counts:** We have included an ablation study showing that the results remain consistent across different numbers of bins (Appendix B).
> - **Sample Size:** We have demonstrated that the dataset size has very little effect on our metrics (Appendix C).
>
> Regarding the concern about **distributional shape**, we are not entirely certain which specific analysis is being requested. We would be grateful for further clarification on this point and are more than willing to conduct the necessary analysis once we have more details.
>
> > 3. The main link—disentanglement as minimal and sufficient representations in the information-bottleneck sense—has been touched on in prior work, so the main novelty of the paper defining concrete evaluation metrics that make these connections concise. I think this is still a meaningful novelty, but the paper might want to present itself more as “a solid and principled evaluation framework” rather than suggesting a completely new theory of disentanglement.
>
> We are not currently aware of specific prior work that explicitly links disentanglement to minimal sufficient representations in the manner we describe. We would be very interested in receiving these references so that we can properly cite and discuss them in our paper.
>
> Regarding the positioning of our contribution, we fully agree that our work should be viewed as a principled evaluation framework rather than a completely new theory of disentanglement. If there are specific sections that suggest a broader theoretical claim, we would appreciate it if you could point them out so we can clarify the text to better reflect the scope of our contribution.
>
> ---
>
> ### Questions
>
> > 1. Can you calrify how siginificant is the conditional independence $p(y_i, y_j | x) = p(y_i| x) p(y_j|x)$?
>
> Please refer to our response to Con 1, where we address the removal of this assumption.
>
> > 2. Possibly a moving-forward question on the possibility for distributed representations, rather than concern: Do you see a way to extend your framework to handle the case where a factor is represented by a group of coordinates rather than a single one?
>
> We agree that extending our framework to handle distributed representations—where factors correspond to groups of coordinates—is an important avenue for future research. We have included a discussion in the "Future Work" section detailing how multivariate estimators, such as KSG or $\mathcal{V}$-Information, could be utilized to measure Minimality and Sufficiency on subspaces rather than single coordinates.
>
> > 3. How sensitive are the Minimality and Sufficiency scores to the binning and sample size? Have you tried different bin counts or alternative MI estimators?
>
> Please refer to our response to Con 2, where we discuss our robustness analysis regarding binning and sample size.
>
> ---
>
> We hope that these responses and the corresponding revisions satisfactorily address the reviewer's concerns. We remain open to any further questions or suggestions.

---

> ### Comment · Action_Editor_t36s · 2025-12-09
>
> Dear Reviewer,
>
> I kindly ask you to copy the comment as an official review. As the authors already reply, you can keep this comment and continue the discussion here, but we need the comment as a formal review in OpenReview to move on to the next stage.
>
> Thank you,
> AE

---

> > ### Comment · Action_Editor_t36s · 2025-12-19
> >
> > Dear Reviewer,
> >
> > I kindly ask you to copy the comment as an official review. As the authors already reply, you can keep this comment and continue the discussion here, but we need the comment as a formal review in OpenReview to move on to the next stage.
> >
> > Thank you, AE

---

### Author Response · Authors · 2025-12-03
**Summary of Revisions (Changes highlighted in Blue)**

We thank the reviewers for their insightful feedback, which has significantly strengthened the paper. We have uploaded a revised manuscript with all major changes highlighted in **blue**.

Key updates include:

1. **Theoretical Refinements:**
    - We corrected the proof and statement of **Theorem 1** (addressing Reviewer vKVM).
    - We removed the assumption regarding the conditional independence of factors given $x$, as we verified it is not required for our main theorems (addressing Reviewer AQT3).
2. **Robustness Analysis:** We added new ablation studies in **Appendix B and C** demonstrating that our metrics are robust to variations in bin counts and sample size (addressing Reviewers 2H2y and AQT3).
3. **Clarifications on Scope & Complexity:** We added **Section 5.3** to discuss computational complexity and practical scaling. We also expanded the **Future Work** section to discuss differentiable optimization and distributed representations.
4. **Experimental Details:** We clarified that **Experiment 6** utilizes real image inputs (not just synthetic data) and explicitly stated the supervision requirements in **Section 4.4** (labels required for factors, but *not* for nuisances).
5. **Definitions & Notation:** We added the formal definition of the interventional robustness score, clarified the term "sub-representations," and corrected notation and citations throughout the manuscript.

---

### Decision · Action_Editor_t36s · 2026-02-13

**Recommendation:** Accept as is

**Audience:**

Yes

**Audience Explanation:**

Disentanglement is still a popular topic in representation learning.

**Claims And Evidence:**

Yes

**Claims Explanation:**

This paper introduces a new methodology to quantify the degree of disentanglement under correlated factors of variations. This is rooted in information theory and the theory of representations. Overall, the paper show convincing experiment on the typical disentanglement data sets and thus I recommend acceptance as is.

As a side note, I'd like to point the authors to Fumero et al., "Leveraging sparse and shared feature activations for disentangled representation learning" where the minimality and sufficiency are also explored for multi-task disentanglement and Yao et al., "Unifying Causal Representation Learning with the Invariance principle" which extends this language to causal representation learning. There may be an interesting connection for the authors there.

Overall, it's a nice paper, and I support acceptance.